# The short-term impact of 3 smoked cannabis preparations versus placebo on PTSD symptoms: A randomized cross-over clinical trial

**Marcel O. Bonn-Miller[1], Sue Sisley[2], Paula Riggs[3], Berra Yazar-Klosinski[4], Julie B. Wang[4], Mallory J. E. Loflin**[5,6]*, **Benjamin Shechet**[2,4], **Colin Hennigan[4], Rebecca Matthews[4], Amy Emerson[4], Rick Doblin[4]**

1 Perelman School of Medicine, University of Pennsylvania, Philadelphia, Pennsylvania, United States of America, 2 Scottsdale Research Institute, Scottsdale, Arizona, United States of America, 3 School of Medicine, University of Colorado, Aurora, Colorado, United States of America, 4 Multidisciplinary Association for Psychedelic Studies, Santa Cruz, California, United States of America, 5 Center of Excellence for Stress and Mental Health & National Center for PTSD, VA San Diego Healthcare System, San Diego, California, United States of America, 6 San Diego School of Medicine, Department of Psychiatry, University of California, San Diego, California, United States of America

* mjloflin@gmail.com

**Data Availability Statement:** All non-identifiable, relevant data are currently attached in the Supporting Information files.

## Abstract

### Importance

There is a pressing need for development of novel pharmacology for the treatment of Post-traumatic Stress Disorder (PTSD). Given increasing use of medical cannabis among US military veterans to self-treat PTSD, there is strong public interest in whether cannabis may be a safe and effective treatment for PTSD.

### Objective

The aim of the present study was to collect preliminary data on the safety and potential efficacy of three active concentrations of smoked cannabis (i.e., High THC = approximately 12% THC and < 0.05% CBD; High CBD = 11% CBD and 0.50% THC; THC+CBD = approximately 7.9% THC and 8.1% CBD, and placebo = < 0.03% THC and < 0.01% CBD) compared to placebo in the treatment of PTSD among military veterans.

### Methods

The study used a double-blind, cross-over design, where participants were randomly assigned to receive three weeks of either active treatment or placebo in Stage 1 (N = 80), and then were re-randomized after a 2-week washout period to receive one of the other three active treatments in Stage 2 (N = 74). The primary outcome measure was change in PTSD symptom severity from baseline to end of treatment in Stage 1.

**Funding:** Authors BY, RD, AE, MB, PR, and SS received Grant Number: RFA#135, an award funded by the Colorado Department of Public Health and Environment (CDPHE): https://www.colorado.gov/pacific/cdphe/approved-medical-marijuana-research-grants The study was also partially funded by the sponsor, The Multidisciplinary Association for Psychedelic Studies (MAPS): https://maps.org/research/mmj/ The sponsor designed the protocol with input from from MB, SS, and PR. The sponsor monitored the data quality, conducted data analysis, contributed to decision to publish, and assisted with preparation of manuscript through critical review.

**Competing interests:** Author MBM is an employee of Canopy Growth Corporation, during which time he has received stock options, serves on the Board of Directors for AusCann Group Holdings Limited, was a prior employee of Zynerba Pharmaceuticals, and has received consulting fees from Tilray Inc. Author ML serves on the scientific advisory board for FSD Pharma and has received consulting fees from Greenwich Biosciences, Zynerba Pharmaceuticals, and Tilray Inc in the past two years. Authors RD, BY, JW, BS, CH, RM, and AE receive salary from the Multidisciplinary Association for Psychedelic Studies (MAPS), a 501 (c)(3) non-profit research and educational organization. Author SS receives salary from the Scottsdale Research Institute, which is a private LLC and has no shareholders. The Academic Editor, BLF, co-authored "The state of clinical outcome assessments for cannabis use disorder clinical trials: A review and research agenda" (https://pubmed.ncbi.nlm.nih.gov/32360455/) with one of the authors, MBM. This article was a result of a meeting where a large number of investigators came together to discuss clinical trial outcomes with representative from NIH and FDA. No other relationship between this author and the Academic Editor exists. This does not alter our adherence to PLOS ONE policies on sharing data and materials.

## Results

The study did not find a significant difference in change in PTSD symptom severity between the active cannabis concentrations and placebo by the end of Stage 1. All three active concentrations of smoked cannabis were generally well tolerated.

## Conclusions and relevance

The present study is the first randomized placebo-controlled trial of smoked cannabis for PTSD. All treatment groups, including placebo, showed good tolerability and significant improvements in PTSD symptoms during three weeks of treatment, but no active treatment statistically outperformed placebo in this brief, preliminary trial. Additional well-controlled and adequately powered studies with cannabis suitable for FDA drug development are needed to determine whether smoked cannabis improves symptoms of PTSD.

## Trial registration

Identifier: NCT02759185; ClinicalTrials.gov.

## Introduction

Posttraumatic Stress Disorder (PTSD) is a serious, worldwide public health problem. In the United States the lifetime prevalence of PTSD in the general population is between 6 and 10% [1,2], and between 13 and 31% in US military veterans [2,3]. PTSD is typically a chronic condition [4,5], and is associated with high rates of psychiatric and medical co-morbidity, disability, suffering, and suicide [4,6–8]. Food and Drug Administration (FDA)-approved pharmacological treatments for PTSD are currently limited to two selective serotonin reuptake inhibitors (SSRIs): sertraline and paroxetine, which have significantly lower effect sizes (SMD between -.28 and -.56) compared to trauma-focused psychotherapy (SMD between -1.01 and -1.35) [9,10]. Indeed, the current Department of Defense (DoD) and Department of Veterans Affairs (VA) best practice guidelines for treatment of PTSD recommend psychotherapy over pharmacotherapy [11]. However, the majority of military veterans with PTSD who receive one of the best practices psychotherapies for PTSD, which were determined efficacious through clinical trials, do not remit or reduce symptoms below clinical thresholds by the end of treatment [12,13].

There is a strong public interest, particularly among Patients with PTSD, clinicians, and researchers, in whether cannabis can be an effective pharmacological treatment option for individuals with PTSD, or a safe alternative treatment for patients who do not respond to current front-line treatment. Cross-sectional and prospective studies document the widespread use of cannabis by individuals with PTSD [14,15]. Moreover, veterans with PTSD who do not show remission following standard treatment are more likely to use cannabis following completion of PTSD treatment [16]. Two recent prospective studies of patients using cannabis to self-treat PTSD provide evidence that whole plant cannabis can produce short [17] and long-term relief of PTSD symptoms [18].

There is some preclinical evidence that at least two of the active compounds in cannabis, delta-9-tetrahydrocannabinol (THC; the primary constituent responsible for intoxication from cannabis) and cannabidiol (CBD; one of the non-intoxicating cannabinoids in cannabis), can positively impact processes that underly PTSD pathology [19]. Specifically, administration

of CBD in rats and mice dampens cue-elicited fear responses [20,21], while administration of THC and THC+CBD appears to block reconsolidation of fear memory [22]. Likewise, both THC and CBD when administered alone facilitate fear extinction learning [23,24], which is a critical component for recovery from PTSD [25,26]. This work suggests that THC and/or CBD could modify how patients with PTSD experience and respond to reminders of trauma.

In addition to cannabis' potential to perhaps modify mechanisms that maintain the core psychopathology of PTSD, early phase clinical data on isolated cannabinoid constituents in humans suggest that active components of cannabis might provide acute relief from specific symptoms of PTSD. For example, two open-label studies and one randomized placebo controlled trial found that administration of low doses of a THC analogue led to improvements in self-reported subjective sleep quality, decreased frequency of nightmares, and improvements in self-reported overall well-being among those with PTSD [27–29].

While these data appear promising, the potential therapeutic effects of smoked, herbal cannabis on PTSD have not been examined in a randomized, placebo controlled trial. Military veterans with PTSD are overwhelmingly choosing smoked cannabis to self-treat PTSD and related conditions [30]. Moreover, herbal cannabis varies significantly across plants in its THC and CBD content [29]. While both cannabinoids could hold therapeutic value, unlike THC, CBD is non-intoxicating and does not carry significant risk of abuse [30]. In addition, CBD may temper the anxiogenic effects of THC in cannabis preparations that contain both CBD and THC [31,32]. It is unclear whether THC, CBD, or some combination of compounds may lead to greater reductions in PTSD symptoms with better safety profiles compared to other combinations. In addition, previous clinical studies rely entirely on standardized dosing, rather than test more naturalistic and generalizable ad libitum dosing regimens. This is a major limitation of previous research because there is substantial individual variability in cannabinoid tolerability [31]. Indeed, military veterans who use cannabis for PTSD tend to self-titrate to much larger doses than those tested in research studies [32,33].

The primary objective of the present study was to conduct a randomized placebo-controlled trial to assess the safety and potential efficacy of smoked, herbal cannabis for the treatment of PTSD in military veterans. Specifically, the study was designed to examine the independent effects of ad libitum use of up to 1.8 grams/day of three active preparations of smoked cannabis: (i) High THC, (ii) High CBD, and (iii) one-to-one ratio of THC and CBD (THC+CBD) against placebo on PTSD symptoms in a sample of veterans with PTSD.

## Methods

### Trial design

The trial protocol can be found at https://maps.org/research-archive/mmj/MJP1-Protocol-Amend4-oct-13-2015.pdf. The study received ethics approval from the Copernicus Group Independent Review Board (IRB) and was conducted in accordance with all local and Federal laws and regulations, including obtaining written informed consent from all study participants. The study included a randomized, double-blind, placebo-controlled, crossover trial of smoked cannabis containing three different concentrations of THC and CBD, and placebo. The cross-over design included two stages with four treatment groups in Stage 1 (High THC, High CBD, THC+CBD, and placebo) and re-randomization into three active treatment groups in Stage 2 (High THC, High CBD, and THC+CBD). The primary aim of the study was to determine whether change in PTSD symptom severity at the end of Stage 1 (primary study endpoint) differed by condition. The crossover design allowed for additional comparisons of within-subject and between-subject differences in safety and preliminary efficacy across the two Stages and allowed for assessment of participants' preference for cannabis concentrations

assigned in either Stage 1 vs. Stage 2. Each stage included three weeks of *ad libitum* use up to 1.8 grams/day of the assigned treatment followed by a two-week cessation period. This upper limit was necessary due to the outpatient setting for self-administration and the Schedule 1 controlled substance status of cannabis.

Primary outcome and safety assessments were conducted at baseline (visit 0), end of treatment in Stage 1 (visit 5; primary study endpoint), following the Stage 1 cessation period/Stage 2 baseline (visit 7), and end of treatment in Stage 2 (visit 12). Self-reported assessment of withdrawal symptoms was conducted at screening, baseline, and weekly during the two-week cessation periods following each stage of treatment (visits 6, 7, 13, 14). Secondary outcomes were assessed throughout the study before/after treatment and cessation periods.

**Participants.** Study participants were recruited using community-based advertisements, presentations, and website advertisements. Study inclusion and exclusion criteria were as follows:

*Inclusion Criteria.* Individuals were eligible for study enrollment if they (1) were a US military veteran, (2) met DSM-5 (APA, 2013) criteria for PTSD with symptoms of at least six months in duration (index trauma did not have to be related to military service), (3) had PTSD of at least moderate severity based on a CAPS-5 score of = >25 at baseline assessment, (4) were at least 18 years of age, (5) reported they were willing and able to abstain from cannabis use two-weeks prior to baseline assessment, which would be verified by urine toxicology screens at screening and baseline, and agreed to abstain from using non-study cannabis during the trial, (6) were stable on any pre-study medications and/or psychotherapy prior to study entry, and (7) agreed to comply with study procedures.

*Exclusion criteria.* Study participants were excluded if they (1) were pregnant, nursing, or of child bearing potential and not practicing effective means of birth control, (2) had a current or past serious mental illness (e.g., personality disorder, psychotic disorder) determined by the SCID-5-RV [34], or self reported a positive family history (first-degree relative) of psychotic or bipolar disorder (3) were determined at high risk for suicide based on the C-SSRS [35], (4) had allergies to cannabis or other contraindication for smoking cannabis, (5) had a current diagnosis or evidence of significant or uncontrolled hematological, endocrine, cerebrovascular, cardiovascular, coronary, pulmonary, gastrointestinal, immunocompromising, or neurological disease, (6) met DSM-5 criteria for moderate-severe Cannabis Use Disorder on the CUDIT-R (= >11), (7) screened positive for any illicit substance other than cannabis during the two-week screening, or (7) were unable to provide informed consent.

**Randomization and blinding.** The Stage 1 randomization list utilized blocks to ensure equal treatment assignments, and the Stage 2 randomization utilized multiple validated randomization lists that re-randomized participants in a blinded manner. The randomization procedure specified that participants would be randomized to treatment conditions using small block randomization in a 1:1:1:1:1 ratio in Stage 1 and then be re-randomized into two of the three active cannabis conditions (THC, CBD, THC+CBD) with a 1:1 ratio in Stage 2. Randomization in Stage 2 excluded the participant's Stage 1 treatment condition. As placebo was not an option in Stage 2, placebo participants were randomized 1:1 between High THC and High CBD, but were not given the option to be randomized to THC + CBD in order to facilitate simpler programming of the web-based randomization system. This two-step randomization resulted in an unbalanced distribution of Stage 2 participants overall across active dose groups. In order to maintain the blind, a central electronic database was utilized for randomization based on validated computer-generated lists.

All study staff (with the exception of the Randomization Monitor and Drug Product Packaging Technician) and participants were blinded to condition assignments. The blind could only be broken for an individual participant if there was a clinically or medically urgent

emergency requiring knowledge of the participant's condition assignment. This emergency unblinding required approval from the site PI and Coordinating Investigator. Likewise, the unblinded Randomization Monitor could provide dose assignment through the electronic randomization system. Randomization information was only available within the web-based randomization system and only viewable by the designated Randomization Monitor.

**Interventions.** Study drug was obtained from the National Institute on Drug Abuse (NIDA). Four concentrations of cannabis from NIDA included: High THC = approximately 12% THC and < 0.05% CBD); High CBD = 11% CBD and 0.50% THC; THC+-CBD = approximately 7.9% THC and 8.1% CBD, and placebo = < 0.03% THC and < 0.01% CBD. Samples of each batch were tested and confirmed for their concentration levels by an independent third-party analytical testing laboratory in Phoenix, Arizona. The independent testing lab found in two separate analyses that the High THC batch was just 9%, with the other batches very close to what was reported by NIDA.

At the beginning of each stage, participants were asked to visit the clinic site for four hours on two successive days and self-administer under supervision of study staff one dose of the cannabis preparation that they were randomly assigned to in that Stage. Vital signs for safety were collected during these visits (i.e., blood pressure, pulse). The study provided participants a total of 37.8 grams (1.8 grams/day)for the three-week *ad libitum* treatment period along with a metal pipe for treatment delivery (smoked). Participants were asked to refrain from using non-study cannabis, and return any remaining study cannabis that was not used each week. When study drug was returned the clinic team weighed the returned cannabis to calculate participants' average use in grams per day during the treatment period in each stage. Participants were asked to refrain from any cannabis use during a two-week cessation period (between stages), then were re-randomized into one of three active treatment groups. All study participants were provided the option to enroll in an open label extension (Stage 3) with the cannabis of their choice in the same amount they returned unused in Stages 1 and 2 so participants had no disincentives to returning unused amounts. The results of Stage 3 are not reported here.

**Demographic measures.** Baseline demographic information included age, sex, race/ethnicity, education, employment status. Other baseline measures included: whether the index trauma was combat-related, body mass index (BMI), risk for sleep apnea (STOP-bang) [36], and risk for cannabis use disorder (CUDIT-R) [37].

**Safety measures.** Adverse Events (AEs) were assessed at baseline, during the introductory session, self-administration session, end of treatment, and before/after cessation in each stage by asking participants to self-report any side effects experienced over the past week. All AEs were coded by Systems Organ Class. The study physician then rated all AEs by severity (mild, moderate, severe) and study relatedness (i.e., possibly related, probably related, not related). AEs rated possibly related and probably related were collapsed into one "related" category.

Additional safety measures included the 15-item Marijuana Withdrawal Checklist (MWC) (Budney et al., 1999) and the Columbia-Suicide Severity Rating Scale (CSSR-S) (Posner et al., 2011). The MWC was administered at screening, baseline, and each week following cessation of Stages 1 and 2 (visits 6, 7, 13, 14). The CSSR-S was self-administered at all study visits.

**Outcome measures.** The primary outcome of the current study was change in PTSD symptom severity from baseline (visit 0) to end of the three-week treatment period in Stage 1 (visit 5) using the Clinician-Administered PTSD Scale for DSM-5 Total Severity Score (CAPS-5) [38]. The CAPS-5 is a semi-structured clinician interview, and is well-validated for determining PTSD diagnoses consistent with the Diagnostic and Statistical Manual of Mental Disorders, Version 5 (DSM-5) and assessing change in symptom severity over time [39]. PTSD diagnosis is based on meeting the DSM-5 symptom cluster criteria (minimum threshold of symptoms with a score $\geq$ 2) with a qualifying criterion A index trauma. The CAPS-5 Total

Severity Score is calculated by summing the total score for each of the four symptom categories to assess past-month PTSD symptoms on a specific traumatic event: intrusion (Category B), Avoidance (Category C), Mood and Cognition (Category D), and Hyperarousal (Category E). CAPS-5 Total Severity scores range from 0–80, where higher scores indicate worse PTSD severity.

Secondary outcome measures included a modified version of the 20-item self-report PTSD Checklist for DSM-5 (PCL-5) [40], which was changed to assess for past week symptoms, the 20-item general depression subscale and 5-item anxiety subscale from the self-report Inventory of Depression and Anxiety Symptoms' (IDAS) [41], the 80-item self-report Inventory of Psychosocial Functioning (IPF) [42], and the 7-item self-report Insomnia Severity Index (ISI) [43]. Secondary outcome measures were collected at baseline (visit 0 and visit 7), self-administration (visit 4 and visit 10), before cessation (visit 6 and visit 13), and after cessation (visit 7 and visit 14) in both Stage 1 and Stage 2. Total and subscale scores were calculated for each measure.

**Other measures.** The validity of study blinding to active or inactive treatment in Stage 1 was assessed by asking participants and clinicians to independently guess whether the participant was randomized to an active (High THC, High CBD, THC+CBD) or inactive (placebo) treatment group at the end of Stage 1. At the end of Stage 2, participants were asked whether they preferred the treatment to which they were assigned in Stage 1 or Stage 2.

Table 1 includes a summary of all assessments by visit.

**Study power.** The primary study aim was to gather preliminary data on the safety and potential efficacy of different cannabis preparations to treat PTSD among veterans. In the absence of published effect sizes for the impact of THC, CBD, or THC+CBD on CAPS-5 scores, the target sample size was chosen to allow detection of an effect size of 0.4 or greater (small to medium effect) based on between group differences in the primary outcome measure (i.e., change in total CAPS-5 severity score from baseline to the end of Stage 1 active treatment phase). Power analysis suggested that 76 completing participants (n = 19 per group) would be needed to detect an effect size of d = 0.4 at 82% power and .05 significance level. Enrollment and randomization continued until 76 participants completed the Stage 1 outcome assessment. Eighty participants were enrolled and 76 partcipants completed Stage 1.

**Statistical analyses.** Descriptive statistics were performed to test the normality of baseline measures on the total study sample and across each treatment group to ensure adequate randomization. Means, medians, and frequencies were calculated, and within-subject and between-group differences were tested for categorical variables using chi-square tests and t-tests or analysis of variance (ANOVA) for continuous variables.

Safety was analyzed by tabulating the frequency, severity, and relatedness to treatment of AEs. A Chi-square test was used to assess for differences in frequency of AEs across groups. An AE was counted once per subject for each assessment period.

The primary outcome was analyzed using ANOVA to test for between-group differences in change in Total PTSD Severity scores from baseline to end of treatment in Stage 1 (CAPS-5 visits 0 and 7). Secondary outcomes were analyzed using a series of additional ANOVAs to test for between-group differences in change scores from baseline to end of treatment for Stage 1 (CAPS-5 visits 0 and 7; secondary measures visits 0 and 6) and Stage 2 (CAPS-5 visits 7 and 12; secondary measures visits 7 and 13). All dependent variables were tested for normality, and summarized by both mean and median values by group. Within-subject change scores were tested for each treatment group using a series of t-tests. Tukey's pairwise comparisons were used to test for group differences in change scores between all pairs of treatment conditions in Stage 1 and Stage 2. Analyses were conducted consistent with an intent-to-treat (ITT) framework, where all available data from randomized participants who received at least one week's

**Table 1. Summary of assessments by visit.**

| | Screen | | Baseline | Stage 1 | | | Cessation 1 | | Stage 2 | | | Cessation 2 | |
|---|---|---|---|---|---|---|---|---|---|---|---|---|---|
| **Visit #** | Screening / Screen | Screen 2 | Enrollment | Introductory Sessions / Visits 1 & 2 | Self-administration / Visits 3 & 4 | Primary Endpoint / Visit 5 | Cessation 1 / Visit 6 | Stage 2 Baseline / Visit 7 | Introductory Sessions / Visits 8 & 9 | Self-administration / Visits 10 & 11 | Secondary Endpoint / Visit 12 | Cessation 2 / Visit 13 | Final Outcome / Visit 14 |
| Collect AEs | | | ✓ | ✓ | ✓ | | ✓ | ✓ | ✓ | ✓ | | ✓ | ✓ |
| Blood Draw | | ✓ | | | | | ✓ | ✓ | | | | ✓ | ✓ |
| Urinalysis | ✓ | ✓ | | ✓ | ✓ | | ✓ | ✓ | ✓ | ✓ | | ✓ | ✓ |
| Urinary Pregnancy | ✓ | ✓ | | ✓ | ✓ | | ✓ | ✓ | ✓ | ✓ | | ✓ | ✓ |
| ECG | ✓ | | | | | | | | | | | | |
| BP & BT | ✓ | | | ✓ | Weekly | | ✓ | | ✓ | Weekly | | ✓ | ✓ |
| Pulse Oximetry | ✓ | | | ✓B | | | | | ✓B | | | | |
| SCID | ✓ | | | | | | | | | | | | |
| CAPS-5 | | ✓ | | | | ✓ | | ✓F | | | ✓ | | ✓F |
| IDAS | | | ✓ | ✓ | ✓ | | ✓ | ✓ | ✓ | ✓ | | ✓ | ✓ |
| ISI | | | ✓ | ✓ | ✓ | | ✓ | ✓ | ✓ | ✓ | | ✓ | ✓ |
| IPF^G | | | ✓ | | | | ✓ | ✓ | | | | ✓ | ✓ |
| PCL-5^E | ✓ | | ✓ | | ✓ | | ✓ | ✓ | | ✓ | | ✓ | ✓ |
| MWC^H | ✓ | | ✓ | | | | ✓ | ✓ | | | | ✓ | ✓ |
| CUDIT-R | ✓ | | | | | | | | | | | | |
| DEQ | | | | ✓A | | | | | ✓A | | | | |
| Daily Diary | | | ✓ | ✓B | ✓B | ✓ | ✓ | ✓ | ✓B | ✓B | ✓ | ✓ | ✓ |
| C-SSRS | ✓ | | ✓ | ✓ | ✓ | | ✓ | ✓ | ✓ | ✓ | | ✓ | ✓ |
| STOP-Bang | ✓ | | | | | | | | | | | | |
| Belief of Condition | | | | | | | ✓ | | | | | ✓ | |
| TLFB | ✓ | | | | | | | | | | | | |
| DDIS | ✓ | | | | | | | | | | | | |

Note. AE = adverse event; ECG = electrocardiogram; BP = blood pressure; SCID = Structured Clinical Interview for DSM-5; CAPS-5 = Clinician Administered PTSD Scale; IDAS = Inventory of Depression and Anxiety; ISI = Insomnia Severity Index; IPF = Inventory of Psychosocial Functioning; PCL-5 = PTSD Checklist for DSM-5; MWC = Marijuana Withdrawal Checklist; CUDIT-R = Cannabis Use Disorder Identification Test Revised; DEQ = Distressing Event Questionnaire; C-SSRS = Columbia Suicide Severity Rating Scale; TLFB = Timeline Follow-back; DDIS = Dissociative Disorders Interview Scale.

A = Completed before & immediately after self-administration and every 30 minutes thereafter until end of session.

B = Completed immediately after each self-administration.

C = Completed one week prior to the first Introductory Session.

D = Final week only.

E = Measured past week. LEC plus Criterion A assessed at baseline only.

F = CAPS-5 assessments at these timepoints measured the last month. All others measured the last week.

G = Visit 6 based on past 3 weeks, Visit 7 based on the past 2 weeks, Visit 13 based on the past 3 weeks, Visit 14 based on the past 2 weeks.

H = Measured the past 2 weeks.

supply of study drug (N = 80) were summarized for baseline characteristics and entered into the models. However, the use of ANOVA tests only allowed for analysis of change in participants who completed outcome assessments (N = 76 for primary outcome analysis).

# Results

## Sample characteristics

A total of 261 individuals completed screening and 51% met eligibility criteria for study inclusion. Eighty participants were enrolled and randomized into one of four treatment groups (n = 20 per group), of which 76 participants completed the Stage 1 outcome assessment. In Stage 2, a total of 74 participants were re-randomized into High THC (n = 29), High CBD (n = 27), or THC+CBD (n = 18). There were no significant Stage 1 treatment assignment differences in demographics or baseline scores on the primary and secondary outcome variables (i.e., CAPS-5, PCL-5, IDAS Social Anxiety, IDAS Depression, IPF, and ISI). Sample demographics and baseline characteristics are summarized in Table 2.

## Treatment adherence and attrition

Rates of engagement and completion are summarized in the Consort Diagram (Fig 1). During Stage 1, 3 participants (3.8%) did not complete endpoint outcome assessments. After Stage 1, 6 participants (7.5%) did not continue into Stage 2. Of the 74 participants who were re-randomized into Stage 2, 3 (4.1%) discontinued treatment due to an AE, and 7 total (9.5%) did not complete Stage 2 endpoint outcome assessments. The overall attrition rate for the percent of randomized participants who dropped out before completing Stage 2 outcome assessments, was 16.3%.

## Cannabis use in grams

In Stage 1, there was no statistically significant difference between groups in total grams of smoked cannabis/placebo during the three-week treatment period (21 days) across the treatment groups (F [3, 71] = 2.23, p = .09). Mean (SD) grams of smoked cannabis/placebo used by each treatment group in Stage 1 were as follows: placebo (M = 8.4, SD = 10.1), High THC (M = 14.6, SD = 10.4), High CBD (M = 14.3, SD = 13.0), THC+CBD (M = 8.2, 6.8).

In stage 2, there was a significant group difference in total grams of smoked cannabis (F [2, 64] = 3.42, p = .04), such that participants in the THC+CBD group used significantly more cannabis (M = 17.6, SD = 10.6), compared to participants randomized to High THC (M = 10.7, SD = 10.9), or High CBD (M = 9.3, SD = 10.5).

## Assessment of study blind

In Stage 1, 60% of placebo participants accurately guessed assignment to an inactive treatment, 58% of High CBD participants accurately guessed that they were in an active condition, and 100% of participants in the High THC and THC+CBD groups accurately guessed assignment into an active treatment condition. Similar results were found for clinicians. In Stage 1, forty-five percent of clinicians accurately guessed placebo participants' assignment in an inactive treatment, 16% accurately guessed High CBD participants' assignment into an active treatment, and 100% accurately guessed that participants assigned to High THC or THC+CBD were randomized into an active treatment. Therefore, the study blind was appropriately upheld only when participants were assigned to High CBD or placebo conditions, but was not upheld when participants were assigned to High THC or High THC/CBD.

**Table 2. Baseline characteristics by treatment group.**

| | TOTAL | | | TREATMENT GROUP | | | | |
|---|---|---|---|---|---|---|---|---|
| | N | Mean, Median, or % | | High THC (N = 20) | High CBD (N = 20) | THC+CBD (N = 20) | Placebo (N = 20) | *p*-value |
| Age, years | | | | | | | | |
| Mean (SD) | 80 | 44.9 (13.8) | | 45.0 (16.6) | 40.4 (11.2) | 50.6 (13.3) | 43.7 (12.5) | .12 |
| Median (IQR) | 80 | 41.2 (21.5) | | 40.3 (25.0) | 37.2 (10.5) | 48.4 (22.2) | 42.2 (20.6) | |
| Min, Max | 80 | 24.3, 77.3 | | 24.3, 77.3 | 24.5, 67.8 | 31.3, 71.1 | 27.2, 68.8 | |
| Sex, N (%) | | | | | | | | |
| Female | 8 | 10.0% | | 1 (5.0%) | 2 (10.0%) | 3 (15.0%) | 2 (10.0%) | .80 |
| Male | 72 | 90.0% | | 19 (95.0%) | 18 (90.0%) | 17 (85.0%) | 18 (90.0%) | .99 |
| Race/Ethnicity, N (%) | | | | | | | | |
| Non-Hispanic White | 53 | 66.3% | | 11 (55.0%) | 14 (70.0%) | 14 (70.0%) | 14 (70.0%) | .92 |
| Other | 27 | 33.7% | | 9 (45.0%) | 6 (30.0%) | 6 (30.0%) | 6 (30.0%) | .80 |
| Education, N (%) | | | | | | | | |
| High School Graduate | 6 | 7.5% | | 0 | 3 (15.0%) | 1 (5.0%) | 2 (10.0%) | .61 |
| Some College/Associate's Degree | 42 | 52.5% | | 13 (65.0%) | 10 (50.0%) | 8 (40.0%) | 11 (55.0%) | .74 |
| College Graduate | 32 | 40.0% | | 7 (35.0%) | 7 (35.0%) | 11 (55.0%) | 7 (35.0%) | .68 |
| Employment, N (%) | | | | | | | | |
| Full- or Part-time | 20 | 29.0% | | 7 (41.2%) | 2 (11.8%) | 4 (25.0%) | 7 (36.8%) | .31 |
| Other | 49 | 71.0% | | 10 (58.8%) | 15 (88.2%) | 12 (65.0%) | 12 (63.2%) | .79 |
| Combat-related (yes), N (%) | 54 | 67.5% | | 13 (65.0%) | 13 (65.0%) | 15 (75.0%) | 13 (65.0%) | .97 |
| Body mass index (BMI) | | | | | | | | |
| Mean (SD) | 80 | 31.9 (7.4) | | 32.0 (8.9) | 31.3 (6.6) | 31.4 (7.4) | 33.1 (7.0) | .88 |
| Median (IQR) | 80 | 30.3 (9.5) | | 29.2 (12.0) | 30.2 (10.5) | 28.9 (7.2) | 33.2 (8.5) | |
| Min, Max | | 19.7, 56.6 | | 20.2, 56.6 | 19.7, 43.9 | 22.1, 49.9 | 20.5, 44.3 | |
| Sleep Apnea (STOP-bang), N (%) | | | | | | | | |
| Low risk = 1 | 22 | 27.5% | | 6 (30.0%) | 6 (30.0%) | 4 (20.0%) | 6 (30.0%) | .91 |
| Intermediate risk = 2 | 12 | 15.0% | | 3 (15.0%) | 3 (15.0%) | 3 (15.0%) | 3 (15.0%) | 1.00 |
| High risk = 3 | 46 | 57.5% | | 11 (55.0%) | 11 (55.0%) | 13 (65.0%) | 11 (55.0%) | .97 |
| CUDIT-R | | | | | | | | |
| Mean (SD) | 80 | 2.6 (2.9) | | 3.9 (3.0) | 2.5 (2.8) | 2.4 (2.8) | 1.7 (2.6) | .10 |
| Median (IQR) | 80 | 2.0 (4.0) | | 4.0 (5.0) | 2.0 (4.5) | 1.5 (4.0) | 0 (2.5) | |
| Min, Max | 80 | 0, 11 | | 0, 11 | 0, 8 | 0, 9 | 0, 10 | |
| | TOTAL | | TREATMENT GROUP | | | | | |
| | N | Mean, Median, or % | | High THC (N = 20) | High CBD (N = 20) | THC+CBD (N = 20) | Placebo (N = 20) | *p*-value |
| Mean (SD) | 80 | 37.2 (7.3) | | 36.6 (7.2) | 36.8 (8.2) | 38.0 (7.8) | 37.3 (6.4) | .50 |
| Median (IQR) | 80 | 37.0 (9.5) | | 36.0 (10.5) | 36.5 (10.5) | 37.0 (11.5) | 37.5 (7.5) | |
| Min, Max | 80 | 25.0, 55.0 | | 26.0, 54.0 | 25.0, 54.0 | 26.0, 55.0 | 26.0, 53.0 | |
| PTSD symptoms, PCL-5 | | | | | | | | |
| Mean (SD) | 79 | 43.7 (15.0) | | 43.6 (12.6) | 42.3 (17.9) | 45.5 (14.7) | 43.6 (15.5) | .93 |
| Median (IQR) | 79 | 45.0 (22.0) | | 45.5 (20.0) | 41.0 (21.0) | 45.0 (20.5) | 48.0 (27.0) | |
| Social Anxiety, IDAS | | | | | | | | |
| Mean (SD) | 80 | 12.0 (3.9) | | 12.0 (4.4) | 12.7 (4.1) | 11.4 (3.0) | 12.1 (4.1) | .77 |
| Median (IQR) | 80 | 11.5 (5.5) | | 11.5 (5.5) | 13.0 (6.5) | 11.5 (4.0) | 11.0 (6.5) | |
| Min, Max | 80 | 5.0, 24.0 | | 6.0, 24.0 | 7.0, 22.0 | 6.0, 19.0 | 5.0, 19.0 | |

*(Continued)*

**Table 2.** (Continued)

| | | | | | | | | |
|---|---|---|---|---|---|---|---|---|
| General Depression, IDAS | | | | | | | | |
| Mean (SD) | 80 | 55.7 (10.3) | | 55.3 (9.5) | 56.6 (11.6) | 56.2 (10.0) | 54.6 (10.8) | .93 |
| Median (IQR) | 80 | 56.0 (14.0) | | 56.5 (12.5) | 56.0 (14.0) | 55.5 (15.0) | 54.5 (15.0) | |
| Min, Max | 80 | 31.0, 77.0 | | 38.0, 73.0 | 31.0, 77.0 | 36.0, 74.0 | 35.0, 70.0 | |
| Psychosocial Functioning, IPF | | | | | | | | |
| Mean (SD) | 80 | 50.7 (8.1) | | 49.8 (8.9) | 53.4 (6.7) | 48.8 (9.4) | 51.0 (6.6) | .31 |
| Median (IQR) [1] | 80 | 51.5 (11.1) | | 52.0 (10.3) | 52.3 (9.7) | 52.3 (10.2) | 50.9 (10.0) | |
| Min, Max | 80 | 22.5, 67.7 | | 22.5, 59.0 | 39.5, 67.7 | 27.3, 62.8 | 39.2, 64.6 | |

NOTES: (1) Abbreviations: CUDIT-R = Cannabis Use Disorders Identification Test-Revised; CAPS-5 = Clinician Administered PTSD Scale; PCL-5 = PTSD Checklist; IDAS = Inventory of Depression and Anxiety; IPF = Inventory of Psychosocial Functioning; (2) STOP-bang = measure of Obstructive Sleep Apnea; (3) BMI categories: < 18.5 = underweight, 18.5 to 24.9 = normal, 25 to 29.9 = overweight, ≥ 30 = obese. (2) Cannabis Used reflects total grams of cannabis use during all three self-administration weeks during each stage for 3 weeks (21 days).

## Treatment preference

At the end of Stage 2, participants who completed final assessments (n = 74) indicated their preference for either their blinded Stage 1 or Stage 2 treatment assignment. Twenty-five participants (34%) indicated a preference for a Stage 1 or Stage 2 assignment to High THC, 10 participants (13%) indicated a preference for a Stage 1 or Stage 2 assignment to High CBD, 26 participants (35%) indicated a preference for a Stage 1 or Stage 2 assignment to THC+CBD, and 4 participants (5%) indicated a preference for a Stage 1 assignment to placebo. Two participants (3%) equally preferred their Stage 1 and Stage 2 treatment assignments.

## Safety outcomes

**Adverse events.** All Adverse Events (AEs) reported during Stage 1 are summarized in Table 3. Number of participants who reported at least one AE did not significantly differ by treatment group in either Stage 1 (p = .38) or Stage 2 (p = .27). Thirty-seven of 60 participants who received THC, CBD, or THC+CBD during Stage 1 (61.7%) reported at least one treatment-related AE by the end of Stage 1. In Stage 2, Forty-five of the 74 participants who received THC, CBD, or THC+CBD (60.8%) reported at least one treatment-related AE during Stage 2. Three of 80 participants (3.8%) reported an unrelated Serious Adverse Event (SAE) during the study, specifically heart palpitations (n = 1; THC+CBD, Stage 1 cessation period), pulmonary embolism (n = 1, High THC, Stage 2), and abscess (n = 1, High CBD, Stage 2). One participant (THC+CBD) discontinued treatment during the introductory session in Stage 1 due to an AE, and two participants discontinued treatment during the introductory session in Stage 2 due to an AE (High CBD and High THC conditions). Across both Stages, 13 total participants terminated from the study early due to an AE (8.4%). The most common AEs reported (i.e., those with >10% frequency) were cough (12.3%), followed by throat irritation (11.7%) and anxiety (10.4%). Emergency unblinding was never used in the study.

One participant who received CBD in Stage 1 (5.0%) reported treatment-related suicidal ideation. One participant from each treatment condition (3.6% - 5.9%) reported treatment-related suicidal ideation in Stage 2.

**Cannabis withdrawal symptoms.** Fig 2 summarizes mean withdrawal symptom scores on the MWC by group at Stage 1 and Stage 2 baseline, end of treatment, and following 1-week of cessation. All treatment groups reported mean withdrawal symptoms in the moderate range (Mean score = 32–38) at baseline assessment (prior to initiating treatment in Stage 1). All

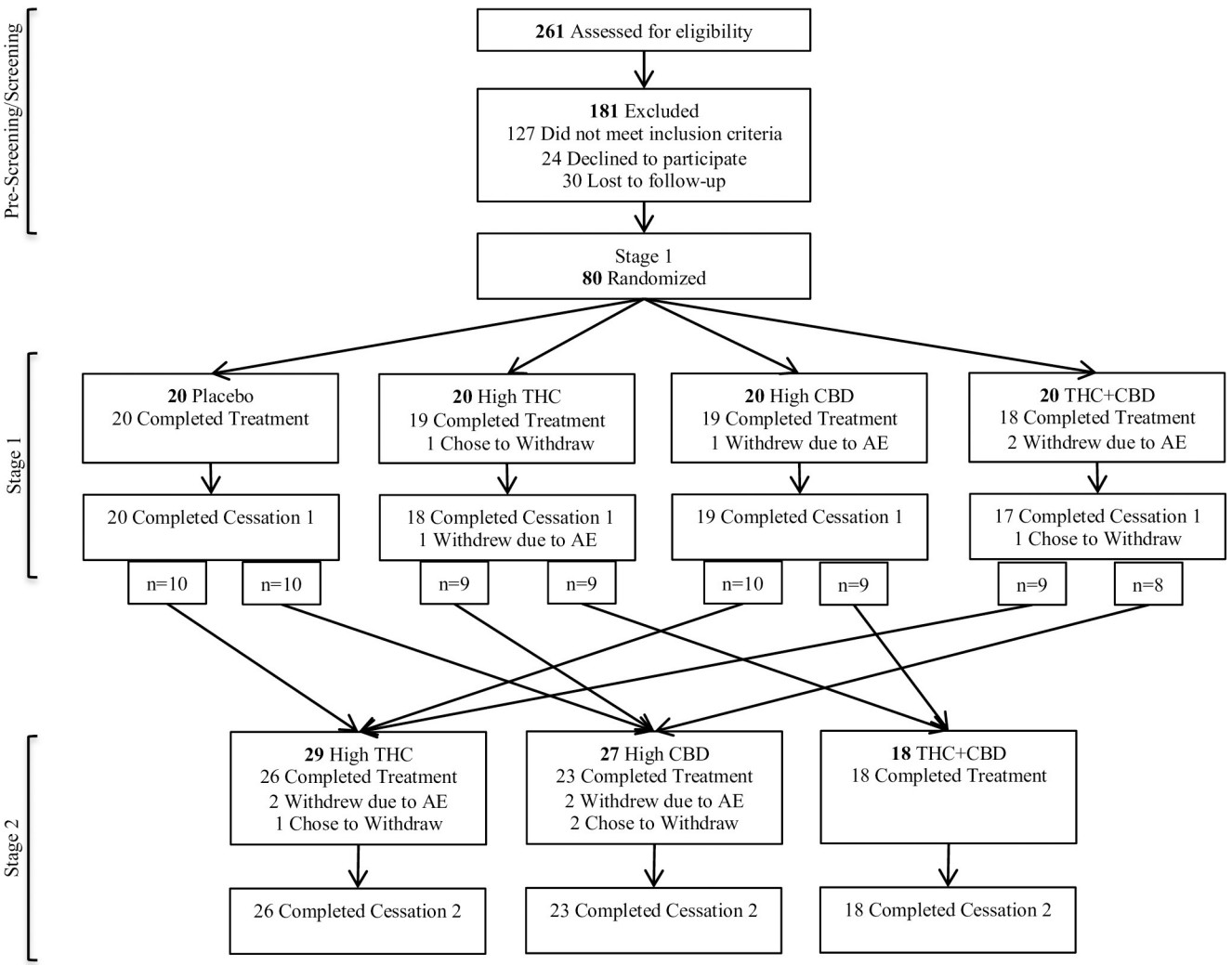

**Fig 1. Consort flow diagram.**

treatment groups showed a significant reduction in withdrawal symptoms from baseline to the end of the treatment phase of Stage 1. Only participants assigned to High THC in Stage 1 reported a significant increase in mean self-reported withdrawal symptoms after one week of cessation from the assigned treatment in Stage 1 ($\Delta$ = 12.6, SD = 11.41, $p$ = .0004). There was no significant change in withdrawal symptoms from the end of Stage 2 treatment to one-week follow-up.

## Primary efficacy outcome

**PTSD symptom severity (CAPS-5).** Results of the analysis of change in total PTSD symptom severity on the CAPS-5 are summarized in Table 4 for both Stage 1 (primary outcome) and Stage 2. In Stage 1, there was no significant between-group difference in CAPS-5 Total Severity scores between treatment groups [$F_{(3, 73)} = 1.85$, $p = .15$]. All four treatment groups, including placebo, achieved significant within-subject reductions in total CAPS-5 Total Severity scores from Stage 1 baseline (visit 0) to end of treatment (visit 5). Specifically, participants who received placebo in Stage 1 reported a mean reduction of 13.1 points (SD = 12.10, p <

**Table 3. Number of participants with adverse events by systems organ class/preferred terms and treatment-relatedness.**

| | STAGE 1 | | | | | | | | STAGE 2 | | | | | |
|---|---|---|---|---|---|---|---|---|---|---|---|---|---|---|
| | Placebo (N = 20) | | High THC (N = 20) | | High CBD (N = 20) | | THC+CBD (N = 20) | | High THC (N = 29) | | High CBD (N = 27) | | THC+CBD (N = 18) | |
| | Related | Not Related | Related | Not Related | Related | Not Related | Related | Not Related | Related | Not Related | Related | Not Related | Related | Not Related |
| **Cardiac disorders** | | | | | | | | | | | | | | |
| Tachycardia | 0 | 0 | 1 (5.0%) | 0 | 0 | 0 | 0 | 0 | 0 | 0 | 0 | 0 | 0 | 0 |
| **Eye disorders** | | | | | | | | | | | | | | |
| Dry eye | 1 (5.0%) | 0 | 2 (10.0%) | 0 | 0 | 0 | 0 | 0 | 1 (3.4%) | 0 | 0 | 0 | 1 (5.6%) | 0 |
| **Gastrointestinal disorders** | | | | | | | | | | | | | | |
| Abdominal discomfort | 0 | 0 | 0 | 0 | 0 | 0 | 1 (5.0%) | 0 | 0 | 0 | 0 | 0 | 0 | 1 (5.6%) |
| Constipation | 0 | 0 | 0 | 0 | 0 | 0 | 0 | 1 (5.0%) | 0 | 0 | 0 | 0 | 0 | 0 |
| Diarrhea | 1 (5.0%) | 1 (5.0%) | 0 | 0 | 0 | 0 | 0 | 1 (5.0%) | 0 | 0 | 0 | 2 (7.4%) | 0 | 1 (5.6%) |
| Dry mouth | 0 | 0 | 3 (15.0%) | 0 | 1 (5.0%) | 0 | 0 | 0 | 2 (6.9%) | 0 | 0 | 0 | 2 (11.1%) | 0 |
| Food poisoning | 0 | 0 | 0 | 0 | 0 | 0 | 0 | 0 | 0 | 0 | 0 | 1 (3.7%) | 0 | 1 (5.6%) |
| Nausea | 0 | 1 (5.0%) | 2 (10.0%) | 0 | 0 | 0 | 5 (25.0%) | 1 (5.0%) | 3 (10.3%) | 0 | 0 | 0 | 1 (5.6%) | 1 (5.6%) |
| Proctalgia | 0 | 1 (5.0%) | 0 | 0 | 0 | 0 | 0 | 0 | 0 | 0 | 0 | 0 | 0 | 0 |
| Toothache | 0 | 0 | 0 | 0 | 0 | 1 (5.0%) | 0 | 0 | 0 | 0 | 0 | 0 | 0 | 0 |
| Vomiting | 0 | 0 | 0 | 0 | 0 | 0 | 1 (5.0%) | 1 (5.0%) | 3 (10.3%) | 0 | 1 (3.7%) | 0 | 1 (3.7%) | 1 (5.6%) |
| **General disorders and administration site conditions** | | | | | | | | | | | | | | |
| Asthenia | 0 | 0 | 0 | 0 | 0 | 0 | 1 (5.0%) | 0 | 0 | 0 | 0 | 0 | 0 | 0 |
| Chest discomfort | 0 | 0 | 0 | 0 | 0 | 0 | 1 (5.0%) | 0 | | 0 | 0 | 0 | | 0 |
| Discomfort | 0 | 0 | 0 | 0 | 0 | 0 | 0 | 0 | 1 (3.4%) | 0 | 0 | 0 | 1 (5.6%) | 0 |
| Fatigue | 0 | 0 | 1 (5.0%) | 0 | 0 | 0 | 0 | 0 | 1 (3.4%) | 0 | 0 | 0 | 0 | 1 (5.6%) |
| Feeling abnormal | 0 | 0 | 0 | 0 | 0 | 0 | 0 | 1 (5.0%) | 1 (3.4%) | 0 | 2 (7.4%) | 0 | 1 (5.6%) | 0 |
| Malaise | 0 | 0 | 0 | 0 | 0 | 1 (5.0%) | 0 | 0 | 0 | 0 | 0 | 1 (3.7%) | 0 | 0 |
| **Infections and infestations** | | | | | | | | | | | | | | |
| Abscess | 0 | 1 (5.0%) | 0 | 0 | 0 | 0 | 0 | 0 | 0 | 0 | 0 | 0 | 0 | 0 |
| Cellulitis | 1 (5.0%) | 0 | 0 | 0 | 0 | 0 | 0 | 0 | 0 | 0 | 0 | 0 | 0 | 0 |
| Ear infection | 0 | 0 | 0 | 0 | 0 | 1 (5.0%) | 0 | 0 | 0 | 0 | 0 | 0 | 0 | 0 |
| Influenza | 0 | 0 | 0 | 1 (5.0%) | 0 | 0 | 0 | 0 | 0 | 0 | 0 | 0 | 0 | 0 |
| | STAGE 1 | | | | | | | | STAGE 2 | | | | | |
| | Placebo (N = 20) | | High THC (N = 20) | | High CBD (N = 20) | | THC+CBD (N = 20) | | High THC (N = 29) | | High CBD (N = 27) | | THC+CBD (N = 18) | |
| | Related | Not Related | Related | Not Related | Related | Not Related | Related | Not Related | Related | Not Related | Related | Not Related | Related | Not Related |
| Respiratory tract infection | 0 | 0 | 1 (5.0%) | 0 | 0 | 0 | 0 | 0 | 0 | 0 | 0 | 0 | 0 | 0 |
| Bacterial infection | 0 | 0 | 0 | 0 | 0 | 0 | 0 | 0 | 0 | 0 | 0 | 1 (3.7%) | 0 | 0 |
| Bacterial vaginosis | 0 | 0 | 0 | 0 | 0 | 0 | 0 | 0 | 0 | 1 (3.4%) | 0 | 0 | 0 | 0 |
| Bronchitis | 0 | 0 | 0 | 0 | 0 | 0 | 0 | 0 | 0 | 0 | 0 | 0 | 0 | 1 (5.6%) |
| Fungal infection | 0 | 0 | 0 | 0 | 0 | 0 | 0 | 0 | 0 | 1 (3.4%) | 0 | 0 | 0 | 0 |
| Sinusitis | 0 | 0 | 0 | 0 | 0 | 0 | 0 | 0 | 0 | 0 | 0 | 2 (7.4%) | 0 | 0 |
| Streptococcal infection | 0 | 0 | 0 | 0 | 0 | 0 | 0 | 0 | 0 | 1 (3.4%) | 0 | 0 | 0 | 0 |
| Upper respiratory tract infection | 0 | 3 (15.0%) | 0 | 2 (10.0%) | 0 | 1 (5.0%) | 0 | 1 (5.0%) | 0 | 2 (6.9%) | 0 | 4 (14.8%) | 0 | 1 (5.6%) |
| **Injury, poisoning and procedural complications** | | | | | | | | | | | | | | |
| Contusion | 0 | 0 | 0 | 1 (5.0%) | 0 | 0 | 0 | 0 | 0 | 0 | 0 | 0 | 0 | 0 |
| Arthropod bite | 0 | 0 | 0 | 0 | 0 | 0 | 0 | 0 | 0 | 1 (3.4%) | 0 | 0 | 0 | 0 |
| Fall | 0 | 0 | 1 (5.0%) | 1 (5.0%) | 0 | 0 | 0 | 0 | 0 | 0 | 0 | 0 | 0 | 1 (5.6%) |
| Foot fracture | 0 | 0 | 0 | 0 | 0 | 0 | 0 | 1 (5.0%) | 0 | 0 | 0 | 0 | 0 | 0 |

*(Continued)*

**Table 3.** (Continued)

| | STAGE 1 Placebo (N = 20) Related | Not Related | High THC (N = 20) Related | Not Related | High CBD (N = 20) Related | Not Related | THC+CBD (N = 20) Related | Not Related | STAGE 2 High THC (N = 29) Related | Not Related | High CBD (N = 27) Related | Not Related | THC+CBD (N = 18) Related | Not Related |
|---|---|---|---|---|---|---|---|---|---|---|---|---|---|---|
| Heat exhaustion | 0 | 0 | 0 | 0 | 0 | 0 | 0 | 0 | 0 | 0 | 0 | 0 | 0 | 2 (11.1%) |
| Joint dislocation | 0 | 0 | 0 | 0 | 0 | 0 | 0 | 0 | 0 | 1 (3.4%) | 0 | 0 | 0 | 0 |
| Poisoning | 0 | 0 | 0 | 0 | 0 | 0 | 0 | 0 | 1 (3.4%) | 0 | 0 | 0 | 0 | 0 |
| **Investigations** | | | | | | | | | | | | | | |
| Weight decreased | 0 | 1 (5.0%) | 0 | 0 | 0 | 0 | 0 | 0 | 0 | 0 | 0 | 0 | 0 | 0 |
| Weight increased | 0 | 0 | 0 | 0 | 0 | 0 | 1 (5.0%) | 0 | 0 | 0 | 0 | 0 | 0 | 0 |
| **Metabolism and nutrition disorders** | | | | | | | | | | | | | | |
| Increased appetite | 0 | 0 | 0 | 0 | 1 (5.0%) | 0 | 1 (5.0%) | 0 | 2 (6.9%) | 0 | 0 | 0 | 0 | 0 |
| **Musculoskeletal and connective tissue disorders** | | | | | | | | | | | | | | |
| Arthritis | 0 | 1 (5.0%) | 0 | 0 | 0 | 0 | 0 | 0 | 0 | 0 | 0 | 0 | 0 | 0 |
| Arthralgia | 0 | 0 | 0 | 0 | 0 | 0 | 0 | 0 | 0 | 1 (3.4%) | 0 | 0 | 0 | 0 |
| Back pain | 0 | 0 | 0 | 2 (10.0%) | 0 | 0 | 0 | 1 (5.0%) | 0 | 2 (6.9%) | 0 | 2 (7.4%) | 0 | 3 (16.7%) |
| Muscle tightness | 0 | 0 | 1 (5.0%) | 0 | 0 | 1 (5.0%) | 0 | 0 | 0 | 0 | 0 | 0 | 0 | 0 |
| Muscle spasms | 0 | 0 | 0 | 0 | 0 | 0 | 0 | 0 | 1 (3.4%) | 0 | 0 | 0 | 0 | 0 |
| Musculoskeletal pain | 0 | 0 | 0 | 0 | 0 | 0 | 0 | 0 | 0 | 1 (3.4%) | 1 (3.7%) | 1 (3.7%) | 0 | 1 (5.6%) |
| Myalgia | 0 | 0 | 0 | 0 | 0 | 1 (5.0%) | 0 | 0 | 0 | 0 | 0 | 0 | 0 | 1 (5.6%) |

| | STAGE 1 | | | | | | | | STAGE 2 | | | | | |
|---|---|---|---|---|---|---|---|---|---|---|---|---|---|---|
| | Placebo (N = 20) | | High THC (N = 20) | | High CBD (N = 20) | | THC+CBD (N = 20) | | High THC (N = 29) | | High CBD (N = 27) | | THC+CBD (N = 18) | |
| | Related | Not Related | Related | Not Related | Related | Not Related | Related | Not Related | Related | Not Related | Related | Not Related | Related | Not Related |
| Neck pain | 0 | 1 (5.0%) | 0 | 0 | 0 | 0 | 0 | 0 | 0 | 0 | 0 | 0 | 0 | 1 (5.6%) |
| Pain in extremity | 0 | 0 | 0 | 0 | 0 | 1 (5.0%) | 0 | 1 (5.0%) | 0 | 0 | 0 | 1 (3.7%) | 0 | 0 |
| Torticollis | 0 | 0 | 0 | 0 | 0 | 1 (5.0%) | 0 | 0 | 0 | 0 | 0 | 0 | 0 | 0 |
| **Nervous system disorders** | | | | | | | | | | | | | | |
| Balance disorder | 1 (5.0%) | 0 | 0 | 0 | 0 | 0 | 0 | 0 | 1 (3.4%) | 0 | 0 | 0 | 0 | 0 |
| Disturbance in attention | 0 | 0 | 1 (5.0%) | 0 | 0 | 0 | 1 (5.0%) | 0 | 2 (6.9%) | 0 | 0 | 0 | 0 | 0 |
| Dizziness | 0 | 0 | 2 (10.0%) | 0 | 0 | 0 | 3 (15.0%) | 0 | 2 (6.9%) | 1 (3.4%) | 0 | 0 | 0 | 1 (5.6%) |
| Head discomfort | 0 | 0 | 0 | 0 | 0 | 0 | 1 (5.0%) | 0 | 0 | 0 | 0 | 0 | 0 | 0 |
| Headache | 4 (20.0%) | 1 (5.0%) | 3 (15.0%) | 0 | 1 (5.0%) | 0 | 3 (15.0%) | 1 (5.0%) | 2 (6.9%) | 2 (6.9%) | 0 | 1 (3.7%) | 1 (5.6%) | 0 |
| Hypoaesthesia | 0 | 0 | 1 (5.0%) | 0 | 0 | 0 | 0 | 0 | 0 | 0 | 0 | 0 | 0 | 1 (5.6%) |
| Migraine | 0 | 0 | 0 | 0 | 0 | 0 | 0 | 0 | 0 | 0 | 0 | 0 | 1 (5.6%) | 0 |
| Memory impairment | 1 (5.0%) | 0 | 1 (5.0%) | 0 | 0 | 0 | 0 | 0 | 0 | 0 | 0 | 0 | 0 | 0 |
| Parosmia | 0 | 0 | 0 | 0 | 0 | 0 | 0 | 0 | 0 | 0 | 0 | 0 | 1 (5.6%) | 0 |
| Sensory disturbance | 0 | 0 | 0 | 0 | 0 | 0 | 1 (5.0%) | 0 | 0 | 0 | 0 | 0 | 0 | 0 |
| Somnolence | 0 | 0 | 3 (15.0%) | 0 | 0 | 0 | 0 | 0 | 1 (3.4%) | 0 | 0 | 0 | 0 | 0 |
| Tremor | 0 | 0 | 0 | 0 | 0 | 1 (5.0%) | 1 (5.0%) | 0 | 0 | 0 | 0 | 0 | 0 | 0 |
| Syncope | 0 | 0 | 0 | 0 | 0 | 0 | 0 | 0 | 0 | 0 | 0 | 1 (3.7%) | 0 | 0 |
| **Psychiatric disorders** | | | | | | | | | | | | | | |
| Aggression | 0 | 0 | 0 | 0 | 0 | 0 | 1 (5.0%) | 0 | 0 | 0 | 0 | 0 | 0 | 0 |
| Affect lability | 0 | 0 | 0 | 0 | 0 | 0 | 0 | 0 | 0 | 0 | 0 | 0 | 1 (5.6%) | 0 |
| Agitation | 0 | 0 | 1 (5.0%) | 0 | 0 | 0 | 0 | 0 | 0 | 0 | 0 | 0 | 1 (5.6%) | 0 |
| Anger | 0 | 0 | 0 | 0 | 0 | 0 | 0 | 0 | 0 | 0 | 0 | 1 (3.7%) | 1 (5.6%) | 0 |
| Anxiety | 1 (5.0%) | 0 | 1 (5.0%) | 0 | 2 (10.0%) | 0 | 2 (10.0%) | 0 | 5 (17.2%) | 1 (3.4%) | 3 (11.1%) | 1 (3.7%) | 2 (11.1%) | 2 (11.1%) |
| Confusional state | 0 | 0 | 0 | 0 | 0 | 0 | 0 | 0 | 0 | 0 | 0 | 0 | 1 (5.6%) | 0 |
| Depression | 0 | 0 | 0 | 0 | 1 (5.0%) | 0 | 0 | 0 | 1 (3.4%) | 0 | 1 (3.7%) | 1 (3.7%) | 0 | 0 |

(*Continued*)

**Table 3.** (Continued)

| | Placebo Related | Placebo Not Related | High THC Related | High THC Not Related | High CBD Related | High CBD Not Related | THC+CBD Related | THC+CBD Not Related | High THC Related | High THC Not Related | High CBD Related | High CBD Not Related | THC+CBD Related | THC+CBD Not Related |
|---|---|---|---|---|---|---|---|---|---|---|---|---|---|---|
| Disorientation | 0 | 0 | 0 | 0 | 0 | 0 | 0 | 0 | 0 | 0 | 1 (3.7%) | 0 | 0 | 0 |

| | STAGE 1 | | | | | | | | STAGE 2 | | | | | |
|---|---|---|---|---|---|---|---|---|---|---|---|---|---|---|
| | Placebo (N = 20) | | High THC (N = 20) | | High CBD (N = 20) | | THC+CBD (N = 20) | | High THC (N = 29) | | High CBD (N = 27) | | THC+CBD (N = 18) | |
| | Related | Not Related | Related | Not Related | Related | Not Related | Related | Not Related | Related | Not Related | Related | Not Related | Related | Not Related |
| Emotional distress | 0 | 0 | 0 | 0 | 0 | 1 (5.0%) | 0 | 0 | 0 | 0 | 0 | 1 (3.7%) | 0 | 0 |
| Hostility | 0 | 0 | 0 | 0 | 0 | 0 | 0 | 0 | 0 | 0 | 0 | 0 | 1 (5.6%) | 0 |
| Hallucinations, visual | 0 | 0 | 1 (5.0%) | 0 | 0 | 0 | 0 | 0 | 0 | 0 | 0 | 0 | 0 | 0 |
| Insomnia | 1 (5.0%) | 0 | 1 (5.0%) | 0 | 0 | 0 | 0 | 0 | 1 (3.4%) | 0 | 2 (7.4%) | 0 | 1 (5.6%) | 0 |
| Irritability | 0 | 0 | 2 (10.0%) | 0 | 1 (5.0%) | 0 | 1 (5.0%) | 0 | 0 | 1 (3.4%) | 1 (3.7%) | 0 | 1 (5.6%) | 2 (11.1%) |
| Mood altered | 0 | 0 | 0 | 0 | 0 | 0 | 0 | 0 | 0 | 0 | 0 | 1 (3.7%) | 0 | 0 |
| Libido increased | 0 | 0 | 0 | 0 | 0 | 0 | 1 (5.0%) | 0 | 0 | 0 | 0 | 0 | 0 | 0 |
| Nightmare | 0 | 0 | 0 | 0 | 1 (5.0%) | 0 | 0 | 0 | 0 | 1 (3.4%) | 0 | 1 (3.7%) | 0 | 0 |
| Obsessive thoughts | 0 | 0 | 0 | 0 | 0 | 0 | 0 | 0 | 0 | 0 | 0 | 1 (3.7%) | 0 | 0 |
| Panic attack | 0 | 1 (5.0%) | 0 | 0 | 0 | 0 | 0 | 0 | 0 | 0 | 0 | 0 | 0 | 1 (5.6%) |
| Paranoia | 0 | 0 | 1 (5.0%) | 0 | 0 | 0 | 4 (20.0%) | 1 (5.0%) | 3 (10.3%) | 0 | 1 (3.7%) | 0 | 1 (5.6%) | 0 |
| Suicidal ideation | - | 0 | 0 | 0 | 0 | 0 | 0 | 0 | 1 (3.4%) | 1 (3.4%) | 1 (3.7%) | 0 | 0 | 0 |
| Restlessness | 0 | 0 | 0 | 0 | 1 (5.0%) | 0 | 0 | 0 | 0 | 0 | 0 | 0 | 0 | 0 |
| **Renal and urinary disorders** | | | | | | | | | | | | | | |
| Pollakiuria | 0 | 1 (5.0%) | 0 | 0 | 0 | 0 | 0 | 0 | 0 | 0 | 1 (3.7%) | 0 | 0 | 0 |
| Urine abnormality | 0 | 1 (5.0%) | 0 | 0 | 0 | 0 | 0 | 0 | 0 | 0 | 0 | 0 | 0 | 0 |
| **Respiratory, thoracic and mediastinal disorders** | | | | | | | | | | | | | | |
| Cough | 2 (10.0%) | 0 | 3 (15.0%) | 0 | 4 (20.0%) | 0 | 2 (10.0%) | 0 | 2 (6.9%) | 0 | 2 (7.4%) | 1 (3.7%) | 4 (22.2%) | 0 |
| Dry throat | 0 | 0 | 0 | 0 | 0 | 0 | 0 | 0 | 0 | 0 | 1 (3.7%) | 0 | 0 | 0 |
| Dyspnoea | 0 | 0 | 0 | 0 | 0 | 0 | 0 | 0 | 1 (3.4%) | 0 | 0 | 0 | 0 | 0 |
| Nasal congestion | 0 | 0 | 0 | 0 | 0 | 1 (5.0%) | 0 | 0 | 1 (3.4%) | 0 | 0 | 0 | 0 | 0 |
| Sinus congestion | 1 (5.0%) | 0 | 0 | 0 | 0 | 0 | 0 | 1 (5.0%) | 0 | 0 | 0 | 0 | 0 | 0 |
| Oropharyngeal pain | 0 | 0 | 0 | 0 | 0 | 0 | 0 | 0 | 0 | 0 | 1 (3.7%) | 1 (3.7%) | 0 | 0 |
| Pulmonary embolism | 0 | 0 | 0 | 0 | 0 | 0 | 0 | 0 | 0 | 1 (3.4%) | 0 | 0 | 0 | 0 |
| Respiratory tract congestion | 0 | 0 | 0 | 0 | 0 | 0 | 0 | 0 | 0 | 0 | 0 | 1 (3.7%) | 0 | 0 |

| | STAGE 1 | | | | | | | | STAGE 2 | | | | | |
|---|---|---|---|---|---|---|---|---|---|---|---|---|---|---|
| | Placebo (N = 20) | | High THC (N = 20) | | High CBD (N = 20) | | THC+CBD (N = 20) | | High THC (N = 29) | | High CBD (N = 27) | | THC+CBD (N = 18) | |
| | Related | Not Related | Related | Not Related | Related | Not Related | Related | Not Related | Related | Not Related | Related | Not Related | Related | Not Related |
| Throat irritation | 6 (30.0%) | 0 | 1 (5.0%) | 0 | 4 (20.0%) | 0 | 3 (15.0%) | 0 | 2 (6.9%) | 0 | 1 (3.7%) | 0 | 1 (5.6%) | 0 |
| Upper respiratory tract congestion | 0 | 0 | 0 | 0 | 1 (5.0%) | 0 | 0 | 0 | 0 | 0 | 0 | 0 | 0 | 1 (5.6%) |
| **Skin and subcutaneous tissue disorders** | | | | | | | | | | | | | | |
| Skin irritation | 0 | 0 | 0 | 0 | 0 | 1 (5.0%) | 0 | 0 | 0 | 0 | 0 | 0 | 0 | 0 |
| Hyperhidrosis | 0 | 0 | 0 | 0 | 0 | 0 | 0 | 0 | 0 | 0 | 1 (3.7%) | 0 | 0 | 0 |
| **Surgical and medical procedures** | | | | | | | | | | | | | | |
| Surgery | 0 | 1 (5.0%) | 0 | 0 | 0 | 0 | 0 | 0 | 0 | 0 | 0 | 0 | 0 | 0 |
| Tooth repair | 0 | 0 | 0 | 0 | 0 | 1 (5.0%) | 0 | 0 | 0 | 0 | 0 | 0 | 0 | 0 |

Note: Counts and frequencies represent number of participants reporting the AE.

(1) list of adverse events with N [3] 1 (2) Related = "possibly" or "probably" related (3) each event counted once per subject (4) "date of last visit" used to include AE counts.

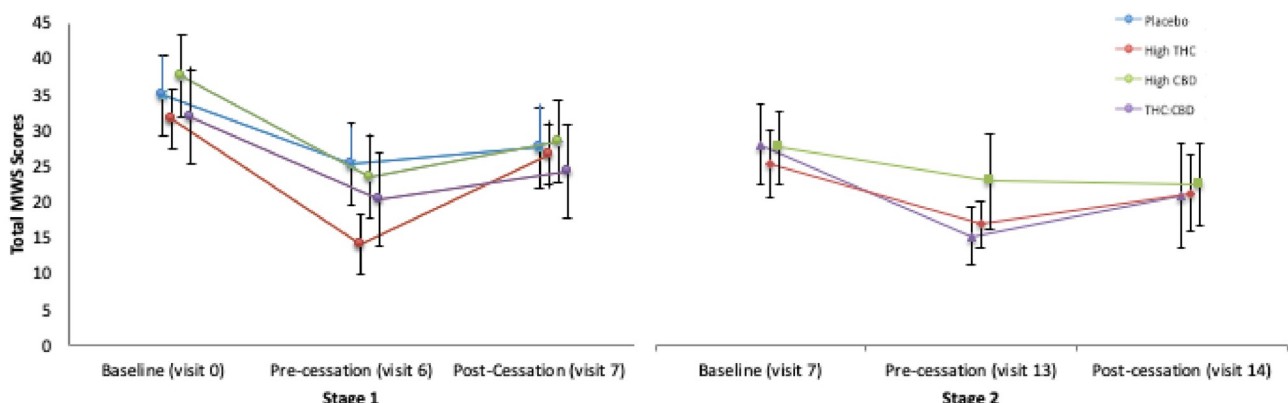

Note. Error bars represent 95% CIs. MWC = Marijuana Withdrawal Checklist; Score of 0 = Average symptom rating of 'none', 15 = Average symptom rating of 'mild'; Score of 30 = Average symptom rating of 'moderate', Score of 45 = Average symptom rating of 'severe'.
[1] Student's t-test to compare pre- to post-cessation scores within group (α = .05); Stage 1 Δ pre- to post-cessation for visit 6 to visit 7; Stage 2 Δ pre- to postcessation
for visit 13 to visit 14
[2] Placebo, Stage 1 Δ = 2.3 (SD = 14.09), p = .59
[3] High THC, Stage 1 Δ = 12.6 (SD = 11.41), p < .001; Stage 2 Δ = 4.3 (SD = 11.50), p = .19
[4] High CBD, Stage 1 Δ = 4.8 (SD = 8.94), p = .26; Stage 2 Δ = 0.41 (SD = 8.38), p = .92
[5] High THC+CBD, Stage 1 Δ = 5.1 (SD = 8.88), p = .39; Stage 2 Δ = 6.3 (SD = 12.18), p = .17

**Fig 2. Mean total marijuana withdrawal scores by treatment condition across Stage 1 & Stage 2.**

.001, d = -1.30), participants who received High THC reported a mean reduction of 15.2 points (SD = 11.3, p < .0001, d = -1.99), High CBD participants reported a mean reduction of 8.4 points (SD = 10.09, p < .05, d = -.79), and THC+CBD participants reported a mean reduction of 8.5 points (SD = 9.88, p < .05, d = -.83).

**Table 4. Mean (SD)/Median (IQR) and analysis of change in CAPS-5 total severity scores by treatment & stage.**

| | | STAGE 1 | | | | | | | | |
|---|---|---|---|---|---|---|---|---|---|---|
| | N | Baseline (Visit 0)[1] | N | Endpoint (Visit 5) | N | Δ | t | p | 95% CI | d |
| Placebo | 20 | 37.3 (6.38)/37.5 (7.5) | 20 | 24.2 (12.79)/26.5 (18.5) | 20 | -13.1 (12.10)/-13.0 (18.0) | -4.10 | .0002* | -19.57, -6.63 | -1.30 |
| High THC | 20 | 36.6 (7.20)/36.0 (10.5) | 19 | 20.5 (8.97)/20.0 (13.0) | 19 | -15.2 (11.03)/-16.0 (13.0) | -6.19 | < .0001* | -21.34, -10.81 | -1.99 |
| High CBD | 20 | 36.8 (8.23)/36.5 (10.5) | 19 | 28.1 (13.26)/29.0 (22.0) | 19 | -8.4 (10.09)/-10.0 (16.0) | -2.47 | .0181* | -15.82, -1.57 | -0.79 |
| THC+CBD | 20 | 38.0 (7.80)/37.0 (11.5) | 19 | 29.6 (12.17)/33.0 (19.0) | 19 | -8.5 (9.88)/-8.0 (12.0) | -2.57 | .0143* | -15.09, -1.65 | -0.83 |
| | | STAGE 2 | | | | | | | | |
| | N | Baseline (Visit 7) | N | Endpoint (Visit 12) | N | Δ | t | p | 95% CI | d |
| High THC | 29 | 25.0 (10.82)/26.0 (12.0) | 26 | 21.4 (12.63)/22.0 (17.0) | 26 | -3.3 (8.22)/-3.5 (9.0) | -1.15 | .2537 | -3.65, -9.99 | -0.31 |
| High CBD | 27 | 28.1 (12.99)/26.0 (19.0) | 23 | 28.2 (11.41)/30.0 (18.0) | 23 | -0.48 (9.07)/-2.0 (16.0) | 0.01 | .9941 | 0.03, -6.99 | 0.01 |
| THC+CBD | 18 | 28.3 (11.11)/27.0 (16.0) | 18 | 16.4 (10.79)/14.5 (13.0) | 18 | -11.8 (12.97)/-8.0 (17.0) | -3.24 | .0027* | -11.83, -19.25 | -1.09 |

[1] d: Calculated effect-sizes of change scores from baseline to endpoint within groups (between subjects) using available data and accounting for differences in sample sizes (Hedge's g).
[2] * = Δ < .05.
[3] Stage 1 between groups analysis of differences: F(3, 73) = 1.85, p = .15 .
[4] Stage 2 between groups analysis of differences: F(2, 64) = 6.92, p = .0019.
In Stage 2, the overall test of between-groups differences in reduction in CAPS-5 Total Severity scores was significant [F(2, 64) = 6.92, p < .01]. Follow-up contrasts showed a significant difference in reductions in CAPS-5 Total Severity scores between participants in the High THC (Δ = -3.3, SD = -8.22, p = .25, d = -.31) and THC+CBD groups (95% CI: 3.82, 18.88, d = -.08), and between participants in the High CBD (Δ = -.48, SD = -9.07, p = .99, d = .01) and THC+CBD groups (95% CI: 1.19, 15.86, d = -1.04).

## Secondary efficacy outcomes

The results of the study's secondary efficacy outcomes are summarized in Table 5.

**Self-reported PTSD symptoms, PCL-5.** In Stage 1, there was no significant difference in PCL-5 change scores between treatment groups from baseline to end of Stage 1.

In Stage 2, mean change in PCL-5 scores significantly differed by treatment condition [$F(2, 63) = 4.06$, $p = .02$]. Specifically, there was a significant difference between High CBD and THC+CBD in PCL-5 change scores, with participants who received THC+CBD reporting greater reductions in PTSD symptoms on the PCL-5 ($\Delta = -16.4$, SD = 16.0, $p < .001$, $d = -1.43$) compared to participants who received High CBD ($\Delta = -9.1$, SD = 11.0, $p = .02$, $d = -.67$).

**IDAS general depression & social anxiety subscales.** In Stage 1, there were no significant differences between treatment conditions in either change in IDAS General Depression or IDAS Social Anxiety scores.

In Stage 2, treatment groups significantly differed in IDAS Social Anxiety mean change scores [$F(2, 63) = -3.08$, $p = .05$] and IDAS General Depression mean change scores [$F(2, 63) = 3.76$, $p = .03$]. Specifically, participants in the THC+CBD condition in Stage 2 reported significant pre-post reductions in IDAS Social Anxiety scores ($\Delta = -2.8$, SD = 3.90, $p = .04$, $d = -.70$), and participants in the High THC ($\Delta = -9.0$, SD = 11.1, $p < .01$, $d = -.90$) and THC+CBD treatment conditions ($\Delta = -13.4$, SD = 10.0, $p < .0001$, $d = -1.68$) reported significant reductions in IDAS General Depression scores in Stage 2.

**ISI insomnia.** In Stage 1, there was no significant difference between treatment conditions in mean change in total insomnia symptoms on the ISI.

In Stage 2, there was no significant difference between treatment groups in mean change scores in total insomnia symptoms on the ISI.

**Psychosocial functioning, IPF.** In Stage 1, there was no significant between-group difference in mean in overall psychosocial functioning (IPF total score).

In Stage 2, there was no significant difference between treatment conditions in IPF mean change scores.

## Posthoc analysis

**CAPS-5 subscale scores B, C, D, and E.** As a follow-up to the primary outcome analysis, posthoc analyses were conducted to test the effects of treatment group on change in each of the primary symptom domains of PTSD (i.e., intrusions, avoidance, negative thoughts and emotions, hyperarousal) using the CAPS-5 B, C, D, and E subscale scores. The posthoc analysis for subscale scores used the same analytic approach as the analysis for primary and secondary outcomes.

**PCL-5, IDAS social anxiety, IDAS general depression, and ISI longitudinal analysis.** Mixed-models for repeated measures (MMRM) were computed to test for group differences over time for all secondary outcome assessments that were measured at more than two time points within each stage. The use of MMRM models allowed all randomized participants' data to be analyzed within the models based on the missing-at-random assumption (MAR).

**Posthoc results.** In Stage 1, there was no significant difference between groups in mean change for any of the subscale scores on the CAPS-5 [Subscale B, $F(3,73) = 1.58$, $p = .20$; Subscale C, $F(3,73) = 1.06$, $p = .37$; Subscale D, $F(3,73) = 2.26$, $p = .09$; Subscale E, $F(3,73) = .84$, $p = .48$].

In Stage 2, there was a significant difference between groups in mean change on the CAPS-5 C (avoidance) [$F(2,64) = 4.95$, $p = .01$] and CAPS-5 D (negative thoughts and emotions) [$F(2,64) = 8.60$, $p < .001$] subscales. Specifically, there was a significant difference between participants who received High CBD and THC+CBD in mean change in CAPS-5 C (avoidance)

**Table 5. Mean (SD)/Median (IQR) and analysis of group change in PCL-5, IDAS social anxiety, IDAS general depression, IPF, and ISI by treatment & stage.**

| | N | STAGE 1 | N | | N | | t | p | | d |
|---|---|---|---|---|---|---|---|---|---|---|
| **PTSD, PCL-5** | N | Pre-drug (Visit 0) | N | Endpoint (Visit 6) | N | $\Delta^1$ | t | p | 95% CI | d |
| Placebo | 19 | 43.6 (15.5)/48.0 (27.0) | 20 | 29.5 (14.9)/30.0 (20.5) | 19 | -14.6 (15.6)/-16.0 (27.0) | -2.89 | .0064* | -23.96, -4.20 | -0.93 |
| High THC | 20 | 43.6 (12.6)/45.5 (20.0) | 19 | 18.8 (9.2)/18.0 (15.0) | 19 | -23.5 (16.5)/-20.0 (29.0) | -6.99 | < .0001* | -24.76, -31.93 | -2.24 |
| High CBD | 20 | 42.3 (17.9)/41.0 (21.0) | 19 | 29.3 (15.1)/31.0 (21.0) | 19 | -12.1 (16.2)/-8.0 (19.0) | -2.43 | .0199* | -12.93, -23.70 | -0.78 |
| THC+CBD | 20 | 45.5 (14.7)/45.0 (20.5) | 18 | 28.9 (15.8)/15.8 (28.0) | 18 | -16.4 (9.1)/-18.0 (9.0) | -3.34 | .0020* | -16.56, -26.62 | -1.09 |
| **Social Anxiety, IDAS** | N | Pre-drug (Visit 0) | N | Endpoint (Visit 6) | N | $\Delta^1$ | t | p | 95% CI | d |
| Placebo | 20 | 12.1 (4.1)/11.0 (6.5) | 20 | 9.7 (3.7)/9.5 (4.5) | 20 | -2.4 (4.3)/-2.5 (4.5) | -1.94 | .0594 | -4.90, 0.10 | -0.62 |
| High THC | 20 | 12.0 (4.4)/11.5 (5.5) | 19 | 8.3 (2.4)/7.0 (3.0) | 19 | -3.7 (4.2)/-2.0 (7.0) | -3.15 | .0033* | -6.06, -1.31 | -1.04 |
| High CBD | 20 | 12.7 (4.1)/13.0 (6.5) | 19 | 9.7 (4.2)/9.0 (6.0) | 19 | -2.7 (2.7)/-3.0 (4.0) | -2.04 | .0483* | -2.74, -5.45 | -0.72 |
| THC+CBD | 20 | 11.4 (3.0)/11.5 (4.0) | 18 | 9.2 (3.7)/8.5 (5.0) | 18 | -2.2 (2.0)/-2.5 (4.0) | -1.99 | .0544 | -2.22, -4.49 | -0.66 |
| **General Depression, IDAS** | N | Pre-drug (Visit 0) | N | Endpoint (Visit 6) | N | $\Delta^1$ | t | p | 95% CI | d |
| Placebo | 20 | 54.6 (10.8)/54.5 (15.0) | 20 | 46.4 (14.2)/44.5 (23.0) | 20 | -8.3 (11.2)/-5.0 (21.5) | -2.07 | .0449 | -16.30, -0.20 | -0.65 |
| High THC | 20 | 55.3 (9.5)/56.5 (12.5) | 19 | 38.4 (10.2)/38.0 (13.0) | 19 | -16.1 (12.6)/-15.0 (10.0) | -5.17 | < .0001* | -22.35, -9.96 | -1.72 |
| High CBD | 20 | 56.6 (11.6)/56.0 (14.0) | 19 | 44.7 (12.9)/42.0 (15.0) | 19 | -11.4 (11.7)/-10.0 (15.0) | -2.86 | .0070* | -19.51, -3.33 | -0.97 |
| THC+CBD | 20 | 56.2 (10.0)/55.5 (10.0) | 18 | 42.6 (11.7)/39.5 (17.0) | 18 | -13.4 (9.1)/-12.0 (11.0) | -3.62 | .0010* | -20.99, -5.89 | -1.26 |
| **Psychosocial Functioning, IPF** | N | Pre-drug (Visit 0) | N | Endpoint (Visit 6) | N | $\Delta^1$ | t | p | 95% CI | d |
| Placebo | 20 | 51.0 (6.6)/50.9 (10.0) | 20 | 50.8 (5.0)/50.7 (4.1) | 20 | -0.2 (6.7)/-1.3 (10.4) | -0.11 | .9106 | -3.97, 3.55 | -0.03 |
| High THC | 20 | 49.8 (8.9)/52.0 (10.3) | 19 | 51.5 (11.3)/54.3 (6.7) | 19 | 1.2 (11.9)/3.3 (8.0) | 0.35 | .7279 | -5.52, 7.83 | 0.17 |
| High CBD | 20 | 53.4 (6.7)/52.3 (9.7) | 19 | 52.7 (6.6)/52.8 (8.8) | 19 | -1.2 (5.5)/-1.6 (6.0) | -0.55 | .5839 | -5.50, 3.15 | -0.11 |
| THC+CBD | 20 | 48.8 (9.4)/52.3 (10.2) | 18 | 52.3 (6.6)/54.1 (8.0) | 18 | 4.1 (8.5)/2.2 (4.5) | 1.51 | .1394 | -1.40, 9.54 | 0.43 |
| **Insomnia, ISI** | N | Pre-drug (Visit 0) | N | Endpoint (Visit 6) | N | $\Delta^1$ | t | p | 95% CI | d |
| Placebo | 20 | 18.4 (5.83)/18.0 (8.0) | 20 | 12.3 (8.58)/12.0 (14.5) | 20 | -6.1 (5.70)/-5.0 (9.5) | -2.63 | .0123* | -10.80, -1.40 | -0.83 |
| High THC | 19 | 17.5 (3.81)/18.0 (7.0) | 19 | 8.1 (5.03)/7.0 (6.0) | 18 | -8.8 (5.20)/-8.5 (7.0) | -6.47 | < .0001* | -12.30, -6.43 | -2.11 |
| High CBD | 20 | 18.1 (4.85)/18.5 (7.5) | 19 | 12.3 (6.71)/12.0 (6.0) | 19 | -5.9 (6.50)/-3.0 (7.0) | -3.13 | .0034* | -9.62, -2.05 | -1.00 |
| THC+CBD | 20 | 18.1 (4.61)/18.0 (6.5) | 18 | 11.9 (5.96)/13.5 (8.0) | 18 | -6.6 (5.18)/-5.5 (5.0) | -3.56 | .0011* | -9.59, -2.62 | -1.17 |
| | | **STAGE 2** | | | | | | | | |
| **PTSD, PCL-5** | N | Pre-drug (Visit 7) | N | Endpoint (Visit 13) | N | $\Delta^1$ | t | p | 95% CI | d |
| High THC | 29 | 30.5 (15.0)/27.0 (19.0) | 26 | 21.8 (10.4)/19.0 (9.0) | 26 | -9.1 (11.0)/-8.0 (16.0) | -2.48 | .0164* | -15.63, -1.79 | -0.67 |
| High CBD | 26 | 36.2 (15.8)/34.5 (24.0) | 22 | 31.0 (19.1)/25.0 (30.0) | 22 | -5.7 (9.3)/-4.0 (16.0) | -1.01 | .3163 | -15.26, 5.24 | -0.30 |
| THC+CBD | 18 | 33.0 (13.2)/31.0 (19.0) | 18 | 16.6 (9.5)/17.0 (12.0) | 18 | -16.4 (16.0)/-18.0 (21.0) | -4.30 | .0001* | -24.22, -8.67 | -1.43 |
| **Social Anxiety, IDAS** | N | Pre-drug (Visit 7) | N | Endpoint (Visit 13) | N | $\Delta^1$ | t | p | 95% CI | d |
| High THC | 29 | 10.4 (4.7)/9.0 (4.0) | 26 | 9.1 (3.2)/8.0 (3.0) | 26 | -1.8 (3.1)/-1.5 (5.0) | -1.25 | .2180 | -3.58, 0.83 | -0.32 |
| High CBD | 26 | 10.2 (3.9)/10.0 (5.0) | 22 | 10.0 (4.8)/9.0 (7.0) | 22 | -.04 (2.7)/0.0 (2.0) | -0.19 | .8492 | -2.79, 2.26 | -0.05 |
| THC+CBD | 18 | 10.1 (5.2)/8.5 (4.0) | 18 | 7.3 (2.2)/7.0 (4.0) | 18 | -2.8 (3.9)/-1.5 (3.0) | -2.10 | .0429* | -5.46, -0.09 | -0.70 |
| **General Depression, IDAS** | N | Pre-drug (Visit 7) | N | Endpoint (Visit 13) | N | $\Delta^1$ | t | p | 95% CI | d |
| High THC | 29 | 48.0 (12.0)/45.0 (16.0) | 26 | 39.0 (7.2)/38.0 (9.0) | 26 | -9.0 (11.1)/-5.0 (15.0) | -3.33 | .0016* | -14.43, -3.57 | -0.90 |
| High CBD | 26 | 50.3 (11.9)/49.0 (19.0) | 22 | 45.8 (14.0)/42.0 (24.0) | 22 | -4.4 (12.4)/-4.5 (21.0) | -1.22 | .2273 | -12.10, 2.95 | -0.35 |
| THC+CBD | 18 | 49.3 (9.3)/48.5 (10.0) | 18 | 35.9 (6.4)/36.5 (10.0) | 18 | -13.4 (10.9)/-11.0 (13.0) | -5.04 | < .0001* | -18.79, -7.99 | -1.68 |
| **Psychosocial Functioning, IPF** | N | Pre-drug (Visit 7) | N | Endpoint (Visit 13) | N | $\Delta^1$ | t | p | 95% CI | d |
| High THC | 29 | 47.8 (9.0)/48.9 (7.9) | 26 | 49.9 (5.9)/50.4 (5.7) | 26 | 0.59 (7.0)/1.3 (9.2) | 1.04 | .3044 | -2.01, 6.32 | 0.27 |
| High CBD | 26 | 50.0 (7.8)/49.1 | 22 | 49.5 (8.8)/49.5 | 22 | 0.28 (4.9)/-0.9 (5.7) | -0.21 | .8373 | -5.33, 4.34 | -0.06 |
| THC+CBD | 18 | (5.7) 50.1 (9.4)/51.1 (5.9) | 18 | (7.3) 51.3 (8.1)/51.9 (10.0) | 18 | 1.2 (8.2)/.39 (11.9) | .41 | .6873 | -4.77, 7.15 | .14 |
| **Insomnia, ISI** | N | Pre-drug (Visit 7) | N | Endpoint (Visit 13) | N | $\Delta^1$ | t | p | 95% CI | d |
| High THC | 29 | 14.5 (5.73)/15.0 (9.0) | 26 | 10.3 (5.66)/9.0 (9.0) | 18 | -4.2 (5.45)/-4.0 (8.0) | -2.74 | .0084* | -7.29, -1.12 | -0.74 |
| High CBD | 26 | 15.5 (7.03)/15.5 (10.00) | 23 | 12.0 (6.95)/10.0 (13.0) | 19 | -3.9 (5.68)/-3.0 (8.0) | -1.73 | .907 | -7.58, 0.58 | -0.50 |

(*Continued*)

**Table 5.** (Continued)

| | | | | | | | | | |
|---|---|---|---|---|---|---|---|---|---|
| THC+CBD | 18 | 13.8 (4.70)/13.5 (9.0) | 18 | 8.3 (5.38)/9.0 (8.0) | 18 | -5.5 (5.66)/-4.5 (9.0) | -3.27 | .0025* | -8.92, -2.08 | -1.09 |

[1] * = α < .05.

[2] *d*: Calculated effect-sizes of change scores from baseline to endpoint within groups (between subjects) using available data and accounting for differences in sample sizes (Hedge's *g*).

[3] PCL-5, Stage 1 between groups analysis of differences: $F(3, 71) = 2.10$, $p = .11$; Stage 2: $F(2, 63) = 4.06$, $p = .02$.

[4] Social Anxiety (IDAS), Stage 1 between groups analysis of differences: $F(3, 72) = .67$ $p = .58$; Stage 2: $F(2, 63) = 3.76$, $p = .03$.

[5] General Depression (IDAS), Stage 1 between groups analysis of differences: $F(3, 72) = 1.67$, $p = .18$; Stage 2: $F(2, 63) = 3.08$, $p = .05$.

[6] Psychosocial Functioning (IPF), Stage 1 between groups analysis of differences: $F(3, 72) = 1.34$, $p = .27$; Stage 2: $F(2, 63) = .09$, $p = .91$.

[7] Insomnia Severity Index (ISI), Stage 1 between groups analysis of differences: $F(3, 71) = 1.00$, $p = .40$; Stage 2: $F(2, 63) = .46$, $p = .63$.

subscale scores (Δ = 1.9, 95% CI: 0.38, 3.35, *d* = -.97) and CAPS-5 D (negative thoughts and emotions) subscale scores (Δ = 5.2, 95% CI: 2.04, 8.26, *d* = -1.13), and between High THC and THC+CBD group participants in mean change in CAPS-5 D (negative thoughts and emotions) subscale scores (Δ = 4.1, 95% CI: 1.09, 7.15, *d* = -1.01). In Stage 2, there were no significant differences between groups in mean change on the CAPS-5 B (intrusions) [F(2,64) = 2.80, p = .07] or CAPS-5 E [F(2,64) = 1.13, p = .33] (hyperarousal) Subscales.

Results of the MMRM analyses testing group differences over time in PCL-5, IDAS Social Anxiety, IDAS General Depression, and ISI appear in Fig 3.

Only IDAS Social Anxiety scores in Stage 2 had a significant time x treatment effect, such that social anxiety showed a quadratic drop in Stage 2 among those in the THC+CBD conditions and those in the High THC and High CBD conditions did not show change over time. All other models failed to find a significant difference between groups over time.

## Discussion

The present study served as the first randomized placebo-controlled trial of smoked cannabis for symptoms of PTSD in US military veterans. Study-related AEs were generally mild to moderate, and did not significantly differ by treatment condition. The study failed, however, to find a significant effect of treatment condition on the primary efficacy outcome, change in total PTSD severity on the CAPS-5 from baseline to end of Stage 1. All treatment groups (placebo, High CBD, High THC, THC+CBD) achieved statistically significant reductions in PTSD severity on the CAPS-5 in Stage 1, with effect sizes for change in mean PTSD severity ranging between d = .83 (High CBD) and d = 1.34 (High THC). These effect sizes are much larger than effect sizes reported for symptom change in other psychopharmacology trials for PTSD. For example, a 2018 meta-analysis reported standardized mean differences between .33 and .97 across PTSD pharmacology trials. The average length of trials reported in the 2018 meta-analysis lasted approximately ten weeks, whereas the current trial's primary endpoint was evaluated after only three weeks of treatment.

The study's failure to detect a significant difference between groups in Stage 1 could perhaps be explained by several confounding factors. First, the study sample included participants with a history of cannabis use. The recruitment of active cannabis users might have increased the potential for biased responding. Given the topical nature of the current trial and its relevance for public policy on medical cannabis, participants might have been biased to report positive effects regardless of condition. Despite many participants already having experience with the drug, nearly half of those receiving placebo believed that they received active cannabis. Prior expectations about cannabis' effects might explain why even those in the placebo

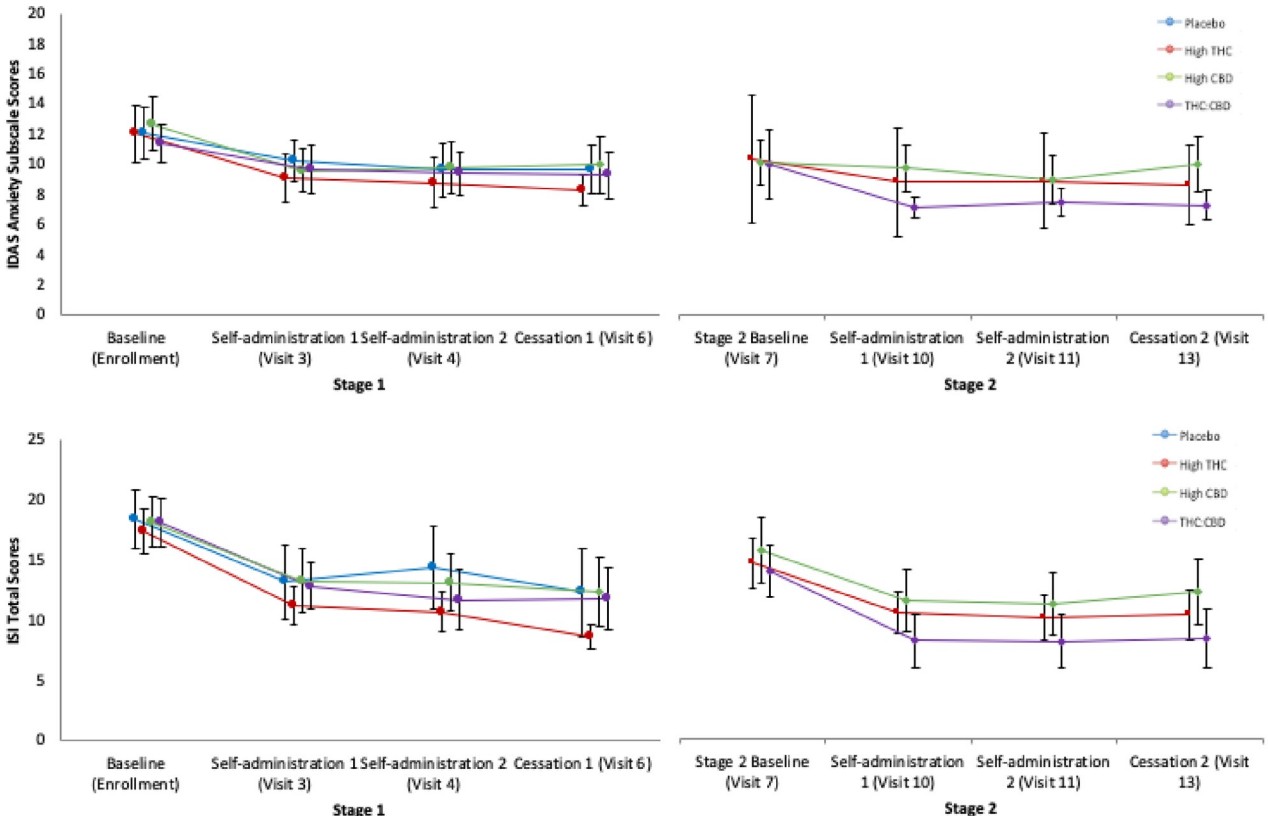

Note. Error bars represent 95% CIs. * = a < .05. PCL-5 = PTSD Checklist for DSM-5 (scores range from 0 to 80); IDAS Depression = Depression Subscale of Inventory of Depression and Anxiety Symptoms (scores range from 0 – 80); IDAS Anxiety = Anxiety Subscale of Inventory of Depression and Anxiety Symptoms (scores range from 0 to 20); ISI = Insomnia Severity Index (scores range from 0 – 28).
[1] PCL-5, Stage 1 group x time interaction: $F_{(9,219)} = 1.52$, p = .14; Stage 2: $F_{(6,194)} = 1.62$, p = .14
[2] General Depression (IDAS), Stage 1 group x time interaction: $F_{(9,220)} =$, p = .05; Stage 2: $F_{(6,194)} = 1.43$, p = .20
[3] Social Anxiety (IDAS), Stage 1 group x time interaction: $F_{(9,219)} = .69$, p = .72; Stage 2: $F_{(6,194)} = 1.84$, p = .09
[4] Insomnia Severity Index (ISI), Stage 1 group x time interaction: $F_{(9,219)} = 1.20$, p = .30; Stage 2: $F_{(6,194)} = .38$, p = .89

**Fig 3. Post hoc Mixed Models for Repeated Measures (MMRM) testing change in PCL-5, IDAS depression, IDAS anxiety, and ISI scores over time.**

condition reported larger than average reductions in PTSD symptoms after only 3 weeks of treatment.

Second, many participants reported significant cannabis withdrawal symptoms at the time of randomization and early in Stage 1. Nearly half of the study sample (43%, n = 34) were positive for THC at study screening and 23% (n = 18) remained positive for THC at Stage 1 baseline, which would suggest chronicity of previous use or continued cannabis use during the two week washout from screening to baseline (lack of compliance with two week abstinence inclusion criteria). Total cannabis withdrawal symptoms averaged in the moderate range for all treatment groups at the start of Stage 1 (despite two weeks of abstinence prior to randomization), then generally reduced to the mild to moderate range by the end of treatment in Stage 1. Participants who received High THC in Stage 1 reported a significant increase in withdrawal following one week of cessation from Stage 1 treatment, which averaged in the moderate range following cessation. While groups did not differ in cannabis withdrawal ratings, the presence of withdrawal and trends in change could confound (or help explain) interpretation of results. We cannot rule out that cessation of cannabis use (in the placebo condition) or reversal of

withdrawal (in the THC and THC+CBD conditions) might have partially been responsible for significant within-subjects change. Moreover, participants randomly assigned to receive High THC in Stage 1 had CUDIT total scores (indicating cannabis use disorder risk) nearly two times greater than participants who were assigned to other active treatment conditions. This is a major confound and limitation of the current study.

Third, total exposure to smoked cannabis was lower than anticipated and might not equate to a full therapeutic dose. In Stage 1, cannabis use in grams ranged from 8.2g (THC+CBD) to 14.6g (high THC) over three weeks (.39g/day to .69g/day on average), despite all participants having access to up to 37.8g over the three week period (1.8g/day). Average cannabis quantity that most cannabis users consume is difficult to estimate from epidemiological studies due to differences in cannabis potency and route of administration. However, large scale studies of medicinal cannabis users treating chronic pain and anxiety report average daily use in ranges closer to 1-3g/day [44,45]. Likewise, two smaller studies that assessed military veterans who use medical cannabis to self-treat PTSD reported median daily use of .85g to 1.14g/day [46] and average use of 3.8g/day [33]. We suspect that participants might have used less cannabis in the current study because of differences between the cannabis available for research trials and the quality of cannabis sold commercially. Several participants spontaneously reported to study staff that the smoke from the cannabis that was provided was "harsher" than they were used to. This difference might also be attributable to the two-week cessation periods mitigating tolerance.

Finally, the present study did not include a placebo arm in Stage 2, which limited the analyses that could be employed in order to take advantage of the crossover design. This significantly limited power to find significant differences across groups. The unexpectedly large response to placebo (d = 1.30), coupled with the small sample size per condition, meant that the current study was underpowered to detect significant differentiation from placebo. If the very large placebo response observed in this study remains consistent in future studies, a trial that tests only one preparation of cannabis (e.g., only high THC cannabis) against placebo would still need a total sample size of nearly one thousand participants (n = 479 per group) to achieve a statistically significant result. However, including a placebo run-in stage to identify and exclude placebo responders in any future trials could substantially reduce this total sample size [47].

Despite these limitations, there were several notable findings. While the study's primary outcome assessment failed to find differentiation from placebo in PTSD symptom change on the CAPS-5, all participants showed a positive response to treatment in a very brief time period. In addition, side effects were general mild and transient. One of the largest concerns from providers regarding self-treatment of PTSD with cannabis is that it may exacerbate PTSD symptoms. While the current study's treatment duration was too brief to identify long-term risk, two-tailed significance tests did not show evidence of symptom exacerbation in any condition. These data provide preliminary evidence of safety of short-term ad libitum cannabis use in this population.

While Stage 2 results should be interpreted with caution given the possible carry-over effects and unbalanced randomization across active dose groups, the study did identify some statistically significant differentiation between groups when particants were re-randomized to only the three active conditions. Given that Stage 2 THC+CBD participants consisted of only those who received active treatment in Stage 1, this cohort might be providing a window into the effects of slightly longer active cannabis treatment. The positive response to treatment evidenced by participants receiving THC+CBD in Stage 2 suggest that it is possible that a longer active treatment period might be necessary to achieve treatment gains that could outperform placebo. It is equally plausible, however, that greater reductions in CAPS-5 severity scores in

Stage 2 among THC and THC+CBD treatment groups were due to attenuation of cannabis withdrawal symptoms, as all treatment groups experienced a 2-week cessation period between Stage 1 and Stage 2. This effect would be consistent with prior literature suggesting that symptoms of cannabis use disorder (CUD) can interfere in successful recovery from PTSD [48].

The current study is unique, in that it trialed whole plant cannabis preparations, rather than single molecule extracts or synthetic pharmaceutical cannabinoids. In addition to reporting change in structured assessments of symptoms, the study results provide critical information about participant preference for cannabinoid preparations when exposed to different whole plant THC and CBD ratios. Consistent with previous work [46], participants in the current study reported a general preference for cannabis types that included significant quantities of THC. This effect might be explained by a signal of efficacy, or simply due to the intoxicating and reinforcing effects of high THC cannabis. Nevertheless, the demand for testing cannabis as a therapeutic for PTSD has largely been driven by military veteran advocacy groups, and yet development of cannabinoid therapeutics has not focused on trialing cannabis preparations that military veterans currently use to self-treat their symptoms. Input from stakeholders on which cannabinoids to trial as cannabis-based medicine for military veteran-specific conditions is certainly justified.

Results from the current trial provide invaluable information for future cannabinoid trials for PTSD. While the null findings raise questions about the utility of continuing to trial whole plant cannabis for the treatment of PTSD, the study found that whole plant, smoked cannabis was generally well tolerated and did not lead to deleterious effects in most participants after 3 weeks of *ad libitum* use. These safety results are consistent with recent real-world evidence studies of whole plant cannabis for PTSD [17,18]. However, those studies both found evidence of potential efficacy as well. Given that many veterans with PTSD are already using cannabis to self-treat their symptoms, identifying which preparations and with which method of administration are most beneficial and/or are least harmful is critical. These future studies will need to take active steps to ensure appropriate blinding despite the intoxicating nature of the drug. While the CBD and placebo conditions were appropriately blinded in the current study, participants and assessors could accurately guess condition when participants received THC-based treatments. Enrolling novice or naïve users, or limiting total THC content to sub-intoxicating doses could improve these blinding issues. Conversely, if higher THC doses are indeed therapeutic, other designs that are less reliant on active blinding, such as randomized withdrawal, might be warranted. Researchers should also consider including objective surrogate endpoint assessments (e.g., physiology, biological specimens, neuroimaging) as secondary indicators of treatment response. As none of these have been validated for a PTSD population as surrogate endpoints, the present study and future studies must utilize the "gold-standard" semi-structured clinical interview for PTSD severity, which is the CAPS-5.

Future studies would also likely benefit from a longer treatment period similar to more traditional medication trials (e.g., 12-weeks), and if including a placebo comparator should plan for a potentially larger than normal placebo response. To mitigate placebo, researchers might consider: 1) including a placebo run-in period prior to randomization to attempt to identify placebo responders, 2) powering the trial for a potentially larger than typical placebo effect, and/or 3) excluding participants with strong apriori beliefs about cannabis' therapeutic effects [e.g., prescreening with expectancy measures, such as Devilly & Borkovec's Credibility/Expectancy Questionnaire (CEQ)]. For generalizability to female veterans and civilians with PTSD, these studies should also attempt to recruit a greater number of female participants, as the current study's sample was overwhelmingly male. Finally, future studies would also greatly benefit from access to high quality flower cannabis. The flower preparations provided through the NIDA drug supply program include all parts of the plant (instead of just buds) and are only

available in specific cannabinoid ratios. Studies that test high quality buds in various phenotypes with variable potencies of cannabinoids and terpene ratios would more closely mirror what is available within state-sponsored medical cannabis programs.

## Conclusions

The present study failed to find a significant group difference between smoked cannabis preparations containing High CBD, High THC, and THC+CBD against placebo in regards to their impact on PTSD symptoms. All treatment groups, including placebo, showed good tolerability and significant improvements in PTSD symptoms during three weeks of treatment. The failure to differentiate treatment groups from placebo is likely attributable to the higher than average treatment response in the placebo condition and to the shorter than average duration of treatment. Higher powered studies that attempt to mitigate the effect of pronounced placebo appear warranted.

## Supporting information

**S1 Checklist.**
(DS_STORE)

**S2 Checklist.**
(DOC)

**S1 File.**
(CSV)

**S2 File.**
(TXT)

**S3 File.**
(TXT)

**S4 File.**
(TXT)

**S5 File.**
(TXT)

**S6 File.**
(TXT)

**S7 File.**
(TXT)

**S8 File.**
(TXT)

**S9 File.**
(TXT)

**S10 File.**
(TXT)

**S11 File.**
(TXT)

**S12 File.**
(XLSX)

**S13 File.**
(PDF)

## Acknowledgments

We thank Scott Hamilton and Ilsa Jerome for their assistance in quality checking all of the study data.

## Author Contributions

**Conceptualization:** Sue Sisley, Paula Riggs, Rick Doblin.

**Data curation:** Berra Yazar-Klosinski, Julie B. Wang.

**Formal analysis:** Berra Yazar-Klosinski, Julie B. Wang.

**Funding acquisition:** Marcel O. Bonn-Miller, Rick Doblin.

**Investigation:** Marcel O. Bonn-Miller, Paula Riggs.

**Methodology:** Marcel O. Bonn-Miller, Paula Riggs, Berra Yazar-Klosinski, Amy Emerson, Rick Doblin.

**Project administration:** Sue Sisley, Benjamin Shechet, Colin Hennigan, Rebecca Matthews, Amy Emerson.

**Resources:** Sue Sisley, Rick Doblin.

**Supervision:** Marcel O. Bonn-Miller, Sue Sisley.

**Validation:** Berra Yazar-Klosinski, Julie B. Wang.

**Visualization:** Berra Yazar-Klosinski, Julie B. Wang.

**Writing – original draft:** Marcel O. Bonn-Miller, Mallory J. E. Loflin.

**Writing – review & editing:** Sue Sisley, Paula Riggs, Berra Yazar-Klosinski, Julie B. Wang, Mallory J. E. Loflin, Benjamin Shechet, Colin Hennigan, Rebecca Matthews, Amy Emerson, Rick Doblin.

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
