## [Decision Letter · Decision Letter 0]

12 Jun 2020

PONE-D-20-03287

The Short-Term Impact of 3 Smoked Cannabis Preparations Versus Placebo on PTSD Symptoms: A Randomized Cross-Over Clinical Trial

PLOS ONE

Dear Dr. Loflin,

Thank you for submitting your manuscript to PLOS ONE. After careful consideration, we feel that it has merit but does not fully meet PLOS ONE’s publication criteria as it currently stands. Therefore, we invite you to submit a revised version of the manuscript that addresses the points raised during the review process.

Firstly let me apologise for the delay in reviewing your manuscript. I have now decided to proceed based on the comments of two reviews. As you can see they concur that the manuscript requires minor revisions.

In your response, please address the issue of blinding, in particular how this might be optimised in future studies of this nature.

We look forward to receiving your revised manuscript.

Kind regards,

Andrew Scholey

Academic Editor

PLOS ONE

Journal Requirements:

2. Thank you for including your competing interests statement; "We have read the journal's policy and the authors of this manuscript have the following competing interests:

Author MM is an employee of Canopy Growth Corporation, during which time he has received stock options, serves on the Board of Directors for AusCann Group Holdings Limited, was a prior employee of Zynerba Pharmaceuticals, and has received consulting fees from Tilray Inc.

Author ML serves on the scientific advisory board for FSD Pharma and has received consulting fees from Greenwich Biosciences, Zynerba Pharmaceuticals, and Tilray Inc in the past two years.

Authors RD, BY, JW, BS, CH, RM, and AE receive salary from the Multidisciplinary Association for Psychedelic Studies (MAPS), a 501(c)(3) non-profit research and educational organization.

Author SS receives salary from the Scottsdale Research Institute, which is private LLC and has no shareholders." 

Reviewers' comments:

Reviewer's Responses to Questions

**Comments to the Author**

1. Is the manuscript technically sound, and do the data support the conclusions?

Reviewer #1: Yes

Reviewer #2: Yes

2. Has the statistical analysis been performed appropriately and rigorously? 

Reviewer #1: Yes

Reviewer #2: N/A

3. Have the authors made all data underlying the findings in their manuscript fully available?

Reviewer #1: No

Reviewer #2: Yes

4. Is the manuscript presented in an intelligible fashion and written in standard English?

Reviewer #1: Yes

Reviewer #2: Yes

5. Review Comments to the Author

Reviewer #1: This study examined the efficacy of 3 formulations of smoked cannabis versus placebo on PTSD symptoms over a three week period. There was no difference between active conditions and placebo on the primary outcome, and all conditions including placebo showed significant improvements in PTSD over a three week period.

This project is justified as there is a serious lack of RCTs on the effects of cannabinoids in PTSD, although pre-clinical and anecdotal evidence are numerous.

Report includes the protocol link, ethics approval, written consent information, detailed statistical analysis section and power analysis.

Major comments-

The findings that there is a high placebo effect is explained by the authors as being due to high expectations or bias by participants on cannabis's efficacy, and by the fact that the placebo blind was upheld. As the authors mention, the use of a 'Beliefs / attitudes toward efficacy of cannabis' measure would have been useful to control for this.

The authors also acknowledge that the lesser quality of cannabis used for the study was likely responsible for the possibly sub-therapeutic amounts consumed. This is a concern for the overall usefulness of the research, given that some participants may have consumed too little of the compounds to elicit any effect. The use of an ad libitum administration approach is potential flaw in this study, although setting a minimum volume to consume may also be problematic.

Given that the participants may have consumed too little of the IP, with possible high expectations of the efficacy of the IP, and all groups showing a similar improvement on PTSD symptoms after a short time, this makes it difficult to draw any meaningful conclusions from the data. These limitations are openly acknowledged and well-discussed, and future studies are clearly necessary to overcome these limitations.

Minor Concerns-

Consider placement of some citations in introduction (after full stops/ commas)

Statistics could be better presented, particularly the secondary outcomes.

Reviewer #2: This study, although in many respects well designed and performed, suffers from difficulties in interpreting the results due complicating factors and the lack of effective blinding. As the authors discuss (although this should have been made clear in the Methods section), the study recruited exclusively active cannabis users, therefore time-varying withdrawal symptoms in the subsequent abstention periods and during the intervention are likely to play a role. Although the study is described as a cross-over, this is not a balanced cross-over design and placebo is lacking in the second stage, therefore the usual analysis methods for crossover studies are not used. Despite a wash-out period, analysing the second stage is complicated by possible carryover effects from the first-stage intervention. The second-stage results are in any case rather incompatible with those of the first stage. The lack of successful blinding in a trial with subjective outcomes (either participant or clinician reported) means that any observed before-after or group comparison effects cannot be unambiguously assigned to real effects of the interventions.

For future trials it would be important to consider whether objective outcomes are available or, failing this, whether the outcome assessors could be more effectively and convincingly blinded.

Minor

1. Describe recruitment methods in the methods section

2. Why were there unequal assignments (n=29/27/18) to arms in stage 2: each arm re-randomised independently?

3. What is the p-value (left side) in Table 2?

4. Treatment preferences are described on p.14, but we need to know which choice of interventions was offered (e.g. so-and-so many preferred High THC to THC+CBD, etc.)

5. I suggest including a table showing the numbers crossing over from each stage 1 intervention to each stage 2 Intervention

6. Tab. 3 needs a legend: state that the numbers in the table refer to numbers of participants, not events.

7. Discussion p.30: why is the first reason given (low cannabis exposure) an explanation for the observed placebo effect?

8. Discussion p.31: interpretation of stage 2 results is unreliable due to possible carry-over effects and to the inconsistencies compared to stage 1.

9. Fig 1: 80, not 261, were randomised

10. Fig 2 & 3. error bars unclear due to overlap – I suggest staggering them by slightly displacing them on the x-axis.

11. Warning: supplementary included datasets are not anonymous: initials and date of birth are included.

6. PLOS authors have the option to publish the peer review history of their article (what does this mean?). If published, this will include your full peer review and any attached files.

Reviewer #1: No

Reviewer #2: Yes: Jeremy Franklin

---

## [Author Response · Author response to Decision Letter 0]

10 Aug 2020

RESPONSE TO REVIEWER FEEDBACK

Thank you for including your competing interests statement. Please confirm that this does not alter your adherence to all PLOS ONE policies on sharing data and materials, by including the following statement: "This does not alter our adherence to PLOS ONE policies on sharing data and materials.” (as detailed online in our guide for authors http://journals.plos.org/plosone/s/competing-interests). If there are restrictions on sharing of data and/or materials, please state these. Please note that we cannot proceed with consideration of your article until this information has been declared.

We have added the statement “This does not alter our adherence to PLOS ONE policies on sharing data and materials” to the disclosures. 

Thank you!

Comments to the Author

Reviewer #1: This study examined the efficacy of 3 formulations of smoked cannabis versus placebo on PTSD symptoms over a three week period. There was no difference between active conditions and placebo on the primary outcome, and all conditions including placebo showed significant improvements in PTSD over a three week period.

This project is justified as there is a serious lack of RCTs on the effects of cannabinoids in PTSD, although pre-clinical and anecdotal evidence are numerous.

Report includes the protocol link, ethics approval, written consent information, detailed statistical analysis section and power analysis.

Major comments-

The findings that there is a high placebo effect is explained by the authors as being due to high expectations or bias by participants on cannabis's efficacy, and by the fact that the placebo blind was upheld. As the authors mention, the use of a 'Beliefs / attitudes toward efficacy of cannabis' measure would have been useful to control for this.

The authors also acknowledge that the lesser quality of cannabis used for the study was likely responsible for the possibly sub-therapeutic amounts consumed. This is a concern for the overall usefulness of the research, given that some participants may have consumed too little of the compounds to elicit any effect. The use of an ad libitum administration approach is potential flaw in this study, although setting a minimum volume to consume may also be problematic.

We completely understand this reviewer’s concerns, as we initially weighed the pros and cons of ad libitum versus standardized dosing as well. Ultimately, the choice of using ad libitum dosing was based on previous literature showing large between subject variability in dose-response for cannabinoids, and data suggesting that participants will self-titrate based on tolerability. In this study, participants were instructed to smoke “ad libitum” with an upper maximum of 1.8 grams/day. This upper limit was necessary due to the outpatient setting for self-administration and the Schedule 1 controlled substance status of the investigational product. In reading this reviewer’s concerns we realized that we never provided our justification for this choice, and have added the following to the introduction: 

“In addition, previous studies rely entirely on standardized dosing, rather than using more naturalistic and generalizable ad libitum dosing regimens. This is a major limitation of previous research because there is substantial individual variability in cannabinoid tolerability[29]. Indeed, military veterans who use cannabis for PTSD tend to self-titrate to much larger doses than those tested in previous trials[30,31].”

Given that the participants may have consumed too little of the IP, with possible high expectations of the efficacy of the IP, and all groups showing a similar improvement on PTSD symptoms after a short time, this makes it difficult to draw any meaningful conclusions from the data. These limitations are openly acknowledged and well-discussed, and future studies are clearly necessary to overcome these limitations.

We agree with this reviewer’s concerns. This was the very first randomized, blinded clinical trial to ever test whole plant cannabis as a treatment for PTSD. The overarching goal was to collect data on safety, determine whether there was a preliminary signal for efficacy in one or all cannabinoid formulations, and collect data on what modifications might be needed in future trials that test whole plant cannabis with appropriate statistical power. While our data cannot definitely tell readers whether cannabis is helpful or harmful for PTSD, we believe that our findings provide the information necessary for designing future trials and are consistent with the goals of an early stage RCT. 

Minor Concerns-

Consider placement of some citations in introduction (after full stops/ commas)

We have updated the references so that they are consistent with the journal style. 

Statistics could be better presented, particularly the secondary outcomes.

To make the results clearer, we only included information in the text of the results section on whether the omnibus between groups tests were significant for all secondary analyses. This helped to reduce redundancy within the results section, as this data is already summarized in the tables, and emphasized to the reader what outcomes were and were not significant between groups. 

Reviewer #2: This study, although in many respects well designed and performed, suffers from difficulties in interpreting the results due complicating factors and the lack of effective blinding. As the authors discuss (although this should have been made clear in the Methods section), the study recruited exclusively active cannabis users, therefore time-varying withdrawal symptoms in the subsequent abstention periods and during the intervention are likely to play a role. Although the study is described as a cross-over, this is not a balanced cross-over design and placebo is lacking in the second stage, therefore the usual analysis methods for crossover studies are not used. Despite a wash-out period, analysing the second stage is complicated by possible carryover effects from the first-stage intervention. The second-stage results are in any case rather incompatible with those of the first stage. The lack of successful blinding in a trial with subjective outcomes (either participant or clinician reported) means that any observed before-after or group comparison effects cannot be unambiguously assigned to real effects of the interventions.

For future trials it would be important to consider whether objective outcomes are available or, failing this, whether the outcome assessors could be more effectively and convincingly blinded.

We are very grateful for this comprehensive feedback. To address the reviewer’s concerns that were not discussed in our Discussion section as limitation, we made the following modifications: 

1) Included information on recruitment methods in the Methods section under Participants (Page 5). 

2) Included suggestions on ways to mitigate the blinding issues in the Discussion section (Page 32)

Minor

1. Describe recruitment methods in the methods section

We have included the following information under Participants (Page 5): 

The study recruited male and female US military veterans with PTSD through community-based advertisements, presentations, and website advertisements. 

2. Why were there unequal assignments (n=29/27/18) to arms in stage 2: each arm re-randomised independently?

This was due to a limitation of the study design. Per protocol, participants were randomized 1:1:1:1 in Stage 1 across four groups. After Stage 1, participants abstained from using product for 2 weeks during Cessation 1, and then they were re-randomized in a blinded manner into 2 of 3 active marijuana dose groups with a 1:1 ratio, while excluding their prior randomized condition in Stage 1. As placebo was not an option in Stage 2, placebo participants were randomized 1:1 between High THC and High CBD, but were not given the option to be randomized to THC + CBD in order to facilitate simpler programming of the web-based randomization system. This two-step randomization resulted in an unbalanced distribution of Stage 2 participants overall across groups, but each individual group had a balanced 1:1 likelihood randomization scheme in Stage 2. Due to anticipated dropouts in the Cessation period following Stage 1, the protocol states: “Due to enrollment and dropouts to achieve N=76 for primary endpoint, the actual number of Stage 2 participants may be higher or lower.” We have included a description of this under Randomization and Blinding methods and updated the Consort Diagram (Figure 1) to show how participants were re-randomized in Stage 2, as well as the accrued subjects and dropouts by each study period. A description of this limitation has also been added to the discussion. 

3. What is the p-value (left side) in Table 2?

This first column was for the total sample, but given that only the baseline differences between groups are relevant for testing we have deleted the first column of p-values to make this clearer. 

4. Treatment preferences are described on p.14, but we need to know which choice of interventions was offered (e.g. so-and-so many preferred High THC to THC+CBD, etc.)

We agree that it would informative to know treatment preferences for each combination of drug. However, we chose not to report the individual treatment preferences of each combination because there were 8 full treatment combinations, with three potential categories of preference for each combination (stage 1 drug, stage 2 drug, or equal preference), meaning that there would be 24 cells for just 74 participants. Therefore, we chose instead to report the percentage of participants who preferred each treatment if it was one of the treatments that they received. 

5. I suggest including a table showing the numbers crossing over from each stage 1 intervention to each stage 2 Intervention

We agree that having a visual representation of this would be helpful and have added this information to Figure 1. 

6. Tab. 3 needs a legend: state that the numbers in the table refer to numbers of participants, not events.

We have added a notes/legend section at the end of the table that states that the counts/frequencies represent the number of participants reporting that AE. 

7. Discussion p.30: why is the first reason given (low cannabis exposure) an explanation for the observed placebo effect?

Upon re-reading, we agree that the ordering of this section doesn’t make sense. We have changed the first paragraph of this section (pg. 30) to now read: 

“The study’s failure to detect a significant difference between groups in Stage 1 could perhaps be explained by several confounding factors. First, the study sample included participants with a history of cannabis use. The recruitment of active cannabis users might have increased the potential for biased responding. Given the topical nature of the current trial and its relevance for public policy on medical cannabis, participants might have been biased to report positive effects regardless of condition. Despite many participants already having experience with the drug, nearly half of those receiving placebo believed that they received active cannabis. Prior expectations about cannabis’ effects might explain why even those in the placebo condition reported larger than average reductions in PTSD symptoms after only 3 weeks of treatment."

8. Discussion p.31: interpretation of stage 2 results is unreliable due to possible carry-over effects and to the inconsistencies compared to stage 1.

We have removed the more thorough discussion of patterns of observed effects in Stage 2 from pg 31, and instead added additional information on safety outcomes, as safety analysis was a primary aim of the study. We also added a sentence to the discussion that all stage 2 results should be interpreted with caution. 

9. Fig 1: 80, not 261, were randomized

We have updated Figure 1 to correctly list that 80 participants were randomized in Stage 1. 

10. Fig 2 & 3. error bars unclear due to overlap – I suggest staggering them by slightly displacing them on the x-axis.

We have updated Figures 2 & 3 so that the error bars between groups do not overlap. 

11. Warning: supplementary included datasets are not anonymous: initials and date of birth are included.

We are very grateful to the reviewer for catching this! All supplementary datasets are now appropriately de-identified.

---

## [Decision Letter · Decision Letter 1]

25 Nov 2020

PONE-D-20-03287R1

The Short-Term Impact of 3 Smoked Cannabis Preparations Versus Placebo on PTSD Symptoms: A Randomized Cross-Over Clinical Trial

PLOS ONE

Dear Dr. Loflin,

Thank you for submitting your manuscript to PLOS ONE. After careful consideration, we feel that it has merit but does not fully meet PLOS ONE’s publication criteria as it currently stands. Therefore, we invite you to submit a revised version of the manuscript that addresses the points raised during the review process.

We look forward to receiving your revised manuscript.

Kind regards,

Bernard Le Foll, M.D., Ph.D.

Academic Editor

PLOS ONE

Reviewers' comments:

Reviewer's Responses to Questions

**Comments to the Author**

1. If the authors have adequately addressed your comments raised in a previous round of review and you feel that this manuscript is now acceptable for publication, you may indicate that here to bypass the “Comments to the Author” section, enter your conflict of interest statement in the “Confidential to Editor” section, and submit your "Accept" recommendation.

Reviewer #1: All comments have been addressed

Reviewer #2: All comments have been addressed

Reviewer #3: (No Response)

2. Is the manuscript technically sound, and do the data support the conclusions?

Reviewer #1: Yes

Reviewer #2: (No Response)

Reviewer #3: Yes

3. Has the statistical analysis been performed appropriately and rigorously? 

Reviewer #1: Yes

Reviewer #2: (No Response)

Reviewer #3: Yes

4. Have the authors made all data underlying the findings in their manuscript fully available?

Reviewer #1: Yes

Reviewer #2: (No Response)

Reviewer #3: Yes

5. Is the manuscript presented in an intelligible fashion and written in standard English?

Reviewer #1: Yes

Reviewer #2: (No Response)

Reviewer #3: Yes

6. Review Comments to the Author

Reviewer #1: I have no further comments, my previous comments have been addressed satisfactorily by the authors.

Reviewer #2: (No Response)

Reviewer #3: This is my first review of this manuscript (now in revised form) describing the outcomes of a double-blind randomized placebo control trial with two stages and three varieties of inhaled whole plant cannabis on PTSD symptoms in a sample of military veterans.

Why was there no placebo control in Stage 2?

Please avoid the use of noun labels (PTSD patients) and replace with the less stigmatizing person first labels (patients with PTSD).

A recent paper has examined the therapeutic effects of inhaled cannabis flower and concentrates on PTSD symptoms using a naturalistic design with ad libitum dosing (see LaFrance et al., 2020; JAD). This recent paper should be acknowledged or the statement that “the potential therapeutic effects of cannabinoids on PTSD have not been examined with smoked, herbal cannabis” should be modified. The statement regarding the failure of previous studies to examine ad libitum dosing could also be slightly modified to reflect this recent contribution to the literature.

If “military veterans who use cannabis for PTSD tend to self-titrate at much larger doses” then the required use of an upper limit may be diminishing the power of the study to detect significant effects. I understand this was required but it should be discussed as a limitation. The use of relatively low potency cannabis could also be offered as a limitation (albeit a necessary one given the limited products produced by the NIDA drug supply). I am particularly intrigued about the finding that the 12% THC product supplied by NIDA tested at a mere 9%!

The sample is predominantly male, and this should be identified as a limitation. Especially in light of recent research (Lafrance et al., 2020) indicating that females may report larger reductions in PTSD symptoms than males after cannabis use and that females are more likely to experience PTSD than males.

Please describe the nature of the 3 SAE reported during Stage 2. Also please define SAE for the reader upon its first instance.

It would seem more appropriate to use mixed factorial ANOVA with symptom severity as the DV, time (baseline, end of treatment) as a within-subjects factor and treatment condition (THC, CBD, THC+CBD, placebo) as a between-subjects factor (rather than relying on change scores for the DV). This might better help to control for/consider variability in symptom severity at baseline (and potentially pull out some trends the authors are detecting). The authors can then report the main effect of time, treatment, and the interaction between these variables. Relatedly, I may be missing something, but it is not clear why MMRM was used for the subscale analyses but not the primary data analyses.

Ideally the direction of treatment effects in Stage 2 would be reported in text as well. This is done nicely for the PCL-5 section but not in the CAPS-5 or Post hoc results sections. While the reader can determine this for themselves using the tables that requires considerably more time and cognitive resources on the part of the reader.

I suggest you cite research showing that chronic cannabis users can test positive for the drug after even a month of abstinence in paragraph 3 of the discussion. Also, in the same paragraph the authors suggest that cessation of cannabis use in the placebo group could have produced the change in symptoms, but this interpretation doesn’t make a lot of sense since scores were decreased in this group and in the cannabis groups. Why would both removing and continued use of a drug produce the same effect?

Participants may have found the NIDA drug “harsher” because NIDA literally uses the whole plant rather than the buds sold in dispensaries. The differences in these products could be further detailed in the discussion.

Minor Issues

There is a typo with the parentheses in the abstract. In the abstract (and Interventions section in the body of the paper) the information about the cannabinoid content of the placebo should be in separate parentheses after “compared to placebo” rather than in the parentheses describing the three active concentrations.

The abstract should contain the sample sizes for each stage.

It would be best to define the acronym DoD/VA

The link to the trial protocol does not work.

There are grammatical issues with these two sentences “The study recruited male and female US military veterans with chronic PTSD through community-based advertisements, presentations, and website advertisements. The study include participants based on the following inclusion and exclusion criteria”

The requirement of being a military veteran appears twice in the inclusion criteria (1 & 4)

It should be stated whether or not emergency unblinding was ever required.

The abbreviations used in the first column of Table 1 should be defined under the table.

“Enrollment and randomization continued until 76 participants completed the Stage 1

outcome assessment (N = 80).” Why is N = 80 and not N = 76 in the parentheses? This confusion appears later in the manuscript as well (Statistical analyses section). I think these issues could be resolved by moving the Sample Characteristics section up to appear after (or within) the Participants section.

The type of effect size used (eta squared, Cohen’s f), the power value used (presumably .80) and alpha level considered should be included in the description of the power analysis.

Given the length of Table 3 the headers should be repeated on each new page.

7. PLOS authors have the option to publish the peer review history of their article (what does this mean?). If published, this will include your full peer review and any attached files.

Reviewer #1: No

Reviewer #2: **Yes: **Jeremy Franklin

Reviewer #3: No

---

## [Author Response · Author response to Decision Letter 1]

11 Jan 2021

Reviewer #3: This is my first review of this manuscript (now in revised form) describing the outcomes of a double-blind randomized placebo control trial with two stages and three varieties of inhaled whole plant cannabis on PTSD symptoms in a sample of military veterans.

Why was there no placebo control in Stage 2?

This was a question that was also raised by previous reviewers and we might not have fully explained the process of how we determined our design. Stage 1 was designed to test our primary efficacy aim and Stage 2 was intended to be entirely exploratory. Unfortunately, we didn’t anticipate that placebo would have a pronounced effect, so we assumed that we would not need a full cross-over analysis to test for differences from placebo. There were also concerns about retention in a clinical drug trial with a population suffering from severe psychiatric symptoms when the intervention wasn’t standard of care. We chose the end of Stage 1 as the primary endpoint of the study to minimize the potential for missing data, assuming that we would see significant drop out between the Stages. The rationale for including a Stage 2 was to allow participants to compare their experience on the two preparations they received in the study to gauge participant preference between the two Stages. 

Please avoid the use of noun labels (PTSD patients) and replace with the less stigmatizing person first labels (patients with PTSD).

Thank you for this suggestion. We have changed this throughout the manuscript. 

A recent paper has examined the therapeutic effects of inhaled cannabis flower and concentrates on PTSD symptoms using a naturalistic design with ad libitum dosing (see LaFrance et al., 2020; JAD). This recent paper should be acknowledged or the statement that “the potential therapeutic effects of cannabinoids on PTSD have not been examined with smoked, herbal cannabis” should be modified. The statement regarding the failure of previous studies to examine ad libitum dosing could also be slightly modified to reflect this recent contribution to the literature.

We thank the reviewer for suggesting the inclusion of this reference. We have added information on the suggested paper into the background and discussion sections of the article and changed the following sentence to now read “…the potential therapeutic effects of smoked, herbal cannabis on PTSD have not been examined in a randomized, placebo controlled trial.”

If “military veterans who use cannabis for PTSD tend to self-titrate at much larger doses” then the required use of an upper limit may be diminishing the power of the study to detect significant effects. I understand this was required but it should be discussed as a limitation. The use of relatively low potency cannabis could also be offered as a limitation (albeit a necessary one given the limited products produced by the NIDA drug supply). I am particularly intrigued about the finding that the 12% THC product supplied by NIDA tested at a mere 9%!

We noted in the discussion section that the NIDA limit might have created an artificial ceiling. However, it is worth noting that while an upper limit was imposed by regulatory bodies, the majority of participants in the study did not use the full amount of cannabis given to them. 

The sample is predominantly male, and this should be identified as a limitation. Especially in light of recent research (Lafrance et al., 2020) indicating that females may report larger reductions in PTSD symptoms than males after cannabis use and that females are more likely to experience PTSD than males.

This is an excellent suggestion. Owing to the population that we were sampling from (military veterans), the sample was predominantly male. We have added this to the limitations section for generalizability to PTSD broadly. 

Please describe the nature of the 3 SAE reported during Stage 2. Also please define SAE for the reader upon its first instance.

We have added the definition of serious adverse event (SAE) in the first instance. We re-ordered some of the information in the AE/SAE summary paragraph because the SAEs actually occurred across both stages. We have added the information on what each SAE was and when in the study it occurred and also updated the Consort figure to make clear where drops/withdraws occurred throughout the trial, as many occurred during the washout periods. 

It would seem more appropriate to use mixed factorial ANOVA with symptom severity as the DV, time (baseline, end of treatment) as a within-subjects factor and treatment condition (THC, CBD, THC+CBD, placebo) as a between-subjects factor (rather than relying on change scores for the DV). This might better help to control for/consider variability in symptom severity at baseline (and potentially pull out some trends the authors are detecting). The authors can then report the main effect of time, treatment, and the interaction between these variables. Relatedly, I may be missing something, but it is not clear why MMRM was used for the subscale analyses but not the primary data analyses.

These are excellent suggestions. Consistent with guidance from FDA, one of the ways that we’re attempting to reduce type 1 error is to adhere to our statistical analysis plan that was finalized before study start. This unfortunately means that we are limited in our ability to adjust our analytic plan for the primary outcome posthoc. 

Regarding the use of MMRM for only a subset of variables, the CAPS was only administered at two timepoints, whereas the PCL, ISI, and IDAS were administered at multiple timepoints. Multiple administration with the PCL, ISI, and IDAS allowed us to use a more robust approach. 

Ideally the direction of treatment effects in Stage 2 would be reported in text as well. This is done nicely for the PCL-5 section but not in the CAPS-5 or Post hoc results sections. While the reader can determine this for themselves using the tables that requires considerably more time and cognitive resources on the part of the reader.

In the last round of review our reviewers suggested that we reduce redundancy by keeping results either in text or in the tables. To best accomplish this, we elected to move the majority of Stage 2 results only to the Tables because Stage 2 was an exploratory stage. We do agree that a brief summary of results for each endpoint would be helpful, though, particularly in explaining any significant results. To balance the feedback from the reviewers, we have added back some of the additional information on Stage 2 results, which summarize the simple main effects that explain all significant omnibus findings. 

I suggest you cite research showing that chronic cannabis users can test positive for the drug after even a month of abstinence in paragraph 3 of the discussion. Also, in the same paragraph the authors suggest that cessation of cannabis use in the placebo group could have produced the change in symptoms, but this interpretation doesn’t make a lot of sense since scores were decreased in this group and in the cannabis groups. Why would both removing and continued use of a drug produce the same effect?

We have added a sentence to paragraph 3 stating that cannabis metabolites can remain in urine for up to 30+ days after cessation of use. 

The reviewer asks an excellent question about one potential limitation that we raised in the manuscript for readers to consider. Since all groups reported at least moderate cannabis withdrawal symptoms at baseline, those randomized to placebo would have experienced continued abatement of their withdrawal symptoms throughout phase 1. This could have been subjectively experienced as alleviation or reduction in distress, which overlaps with psychological symptom measures. Likewise, participants randomized to active THC could have also experienced a reduction in withdrawal through drug exposure. In both scenarios, the previous withdrawal symptoms could potentially confound or overlap with reported reductions in PTSD symptoms. 

Participants may have found the NIDA drug “harsher” because NIDA literally uses the whole plant rather than the buds sold in dispensaries. The differences in these products could be further detailed in the discussion.

We appreciate this comment! We have added a more detailed discussion of the potential issues with using the NIDA cannabis supply in clinical trials. 

Minor Issues

There is a typo with the parentheses in the abstract. In the abstract (and Interventions section in the body of the paper) the information about the cannabinoid content of the placebo should be in separate parentheses after “compared to placebo” rather than in the parentheses describing the three active concentrations.

We appreciate the catch. This has been fixed. 

The abstract should contain the sample sizes for each stage.

We have added the sample sizes for each stage to the abstract. 

It would be best to define the acronym DoD/VA

We have added the description for Department of Defense (DoD) and Veterans Affairs (VA)

The link to the trial protocol does not work.

We have updated the link: https://maps.org/research-archive/mmj/MJP1-Protocol-Amend4-oct-13-2015.pdf

There are grammatical issues with these two sentences “The study recruited male and female US military veterans with chronic PTSD through community-based advertisements, presentations, and website advertisements. The study include participants based on the following inclusion and exclusion criteria”

These sentences now read “Study participants were recruited using community-based advertisements, presentations, and website advertisements. Study inclusion and exclusion criteria were as follows:”

The requirement of being a military veteran appears twice in the inclusion criteria (1 & 4)

This has been corrected. 

It should be stated whether or not emergency unblinding was ever required.

Emergency unblinding was never required. This has been added to the paragraph on AE descriptions. 

The abbreviations used in the first column of Table 1 should be defined under the table.

We have added the abbreviations with their definitions to the key at the bottom of Table 1. 

“Enrollment and randomization continued until 76 participants completed the Stage 1

outcome assessment (N = 80).” Why is N = 80 and not N = 76 in the parentheses? This confusion appears later in the manuscript as well (Statistical analyses section). I think these issues could be resolved by moving the Sample Characteristics section up to appear after (or within) the Participants section.

We appreciate that the reviewer noticed this and identified and fixed the inconsistencies of reporting N throughout the paper. Specifically, we changed the N to 76 in the parengtheses after the sentence that the reviewer noted because only 76 participants completed outcome assessments at the end of Stage 1. We also added columns to all results tables for the N associated with the completed data that were analyzed in the models. We believe that this will make it clearer for the reader to interpret how many participants completed baseline assessments and whether this N was different than the number of assessments that were analyzed. 

The type of effect size used (eta squared, Cohen’s f), the power value used (presumably .80) and alpha level considered should be included in the description of the power analysis.

We have added the following information to the study power section on page 10: 

Power analysis suggested that 76 completing participants (n = 19 per group) would be needed to detect an effect size of d = 0.4 at 82% power and .05 significance level. Enrollment and randomization continued until 76 participants completed the Stage 1 outcome assessment. Eighty participants were enrolled and 76 partcipants completed Stage 1.

Given the length of Table 3 the headers should be repeated on each new page.

Headers have been added to the top of each page of Table 3.

---

## [Editor Report · Decision Letter 2]

1 Feb 2021

The Short-Term Impact of 3 Smoked Cannabis Preparations Versus Placebo on PTSD Symptoms: A Randomized Cross-Over Clinical Trial

PONE-D-20-03287R2

Dear Dr. Loflin,

We’re pleased to inform you that your manuscript has been judged scientifically suitable for publication and will be formally accepted for publication once it meets all outstanding technical requirements.

Kind regards,

Bernard Le Foll, M.D., Ph.D.

Academic Editor

PLOS ONE
---

## [Editor Report · Acceptance letter]

17 Feb 2021

PONE-D-20-03287R2 

The Short-Term Impact of 3 Smoked Cannabis Preparations Versus Placebo on PTSD Symptoms: A Randomized Cross-Over Clinical Trial 

Dear Dr. Loflin:

I'm pleased to inform you that your manuscript has been deemed suitable for publication in PLOS ONE. Congratulations! Your manuscript is now with our production department. 

Kind regards, 

on behalf of

Dr. Bernard Le Foll 

Academic Editor

PLOS ONE